# Response to In Vitro Micropropagation of Plants with Different Degrees of Variegation of the Commercial *Gymnocalycium* cv. Fancy (Cactaceae)

**DOI:** 10.3390/plants14071091

**Published:** 2025-04-01

**Authors:** Carles Cortés-Olmos, Vladimir Marín Guerra-Sandoval, Carla Guijarro-Real, Benito Pineda, Ana Fita, Adrián Rodríguez-Burruezo

**Affiliations:** 1Instituto Universitario de Conservación y Mejora de la Agrodiversidad Valenciana (COMAV), Universitat Politècnica de València (UPV), Camino de Vera s/n, 46022 Valencia, Spain; carlescortes3@gmail.com (C.C.-O.); vlaguesa@posgrado.upv.es (V.M.G.-S.); anfifer@btc.upv.es (A.F.); adrodbur@upvnet.upv.es (A.R.-B.); 2Department Biotechnology and Plant Biology, Escuela Técnica Superior de Ingeniería Agronómica, Alimentaria y de Biosistemas ETSIAAB, Universidad Politécnica de Madrid, Av. Puerta de Hierro, 2, Moncloa-Aravaca, 28040 Madrid, Spain; carla.guijarro.real@upm.es; 3IBMCP, Universitat Politècnica de València (UPV), Camino de Vera s/n, 46022 Valencia, Spain

**Keywords:** citokinins, colored cactus, organogenesis, plant growth regulators, cactus areolas, explant

## Abstract

This study aims to establish efficient in vitro propagation protocols for *Gymnocalycium* cv. Fancy, an ornamental cactus with variegated variants, by evaluating the effects of cytokinin type and explant source on the organogenic response. Plants with different degrees of variegation (0–100%) were classified by size to obtain different explant types (apices, central discs, epicotyls, and hypocotyls). The effects of 6-Benzylaminopurine (BAP, 8 µM), Kinetin (KIN, 4 µM), and Thidiazuron (TDZ, 1 µM) were assessed on shoot production, callus formation, and rhizogenesis. Additionally, we studied the relationship between initial plant variegation and the productivity of the variegated shoots. The best shoot production results were obtained for central discs treated with 1 µM TDZ. Furthermore, a correlation was observed between the activated areole type (green, mixed, or fully colored) and shoot color percentage, enabling precise explant selection. The appearance of differently colored shoots confirms the potential for selecting new lines from this cultivar too. These findings hold significant potential not only for the breeding and propagation of ornamental cacti but also for the cultivation of other edible cacti and their relatives.

## 1. Introduction

Variegated plants represent a significant part of the ornamental plant market due to their aesthetic appearance [1]. This variegation results from a partial or total deficit of chlorophyll in certain regions of the plant, leading to a coloration ranging from yellowish to whitish. The presence of leaves and/or stems with different colors or coloring patterns than those typical of the original species gives variegated plants greater visual beauty and ornamental value, making them highly valued among both amateur and professional gardeners or floriculturists [2,3]. Although these plants tend to be less vigorous than non-variegated ones [4], commercial nurseries work intensively to obtain plants with new colorations or patterns to introduce annually into the market. However, the micropropagation of variegated plants presents specific challenges that have not been fully addressed in the scientific literature, particularly for ornamental cacti species.

Certain cactus cultivars developed from Gymnocalycium mihanovichii (Frič and Gürke) Britton and Rose are commercially important due to their diverse coloration [5] (Figure 1). Many of these variants are clonally propagated through grafting [6,7], a technique widely used for the mass propagation of cacti. Grafting offers several advantages over growing plants from their roots, including enhanced plant development, increased shoot production, accelerated blooming, and intensified flowering [8,9]. Additionally, rootstocks are usually more vigorous and resistant to humidity, pests, and diseases, minimizing losses from rot and simplifying their cultivation [8,9].

Nevertheless, grafting may alter the natural morphology of plants, leading to an atypical appearance that may not appeal to collectors who prefer visually natural plants capable of growing on their own roots. Completely achlorophyllous plants must remain grafted due to their inability to photosynthesize [4,10], making them generally unpopular among collectors. This situation underscores the need for protocols to optimize the production processes of cacti with specific percentages of colorations that are able to develop on their own roots.

In this context, in vitro cultivation can play a determinant role in addressing these challenges, as the regenerative capacity of different structures (mainly areoles) has been observed in various cactus species [7,11,12,13,14,15]. However, most studies have focused on edible species of interest to the food industry, such as pitahaya and prickly pear [13,14,15,16,17,18,19,20,21,22,23]. Consequently, there is a significant gap in knowledge regarding many ornamental species, particularly when propagating variegated individuals or those with particular color patterns, as in most cases, the cellular mechanisms that cause them are unknown [24].

The present study aims to establish an efficient in vitro propagation protocol for *Gymnocalycium* cv. Fancy, assessing the influence of different variegation levels (Figure 2) on shoot formation and growth regulator responses. This commercial hybrid, developed by Cactusloft O.E. (Cullera, Valencia, Spain), originated from controlled crosses between *Gymnocalycium mihanovichii* and *Gymnocalycium fiedrichi* (Werdermann) Pažout, resulting in progeny with diverse morphologies and colorations (Figure 2). Specifically, this research evaluates the organogenic response and in vitro behavior of diverse explants (apical, central disc, hypocotyl, and epicotyl) from plants of different sizes (small, medium, and large) under three previously tested plant growth regulators (PGRs) used on chlorophyll-containing plants of the same cultivar [25]. The study seeks to determine the effect of these plant growth regulators on variegated plants and the relationship between the initial plant variegation proportion and the productivity of shoots with varying degrees of variegation.

Thus, this study focuses on the in vitro responses of *Gymnocalycium* plants with varying degrees of variegation (Figure 2). Therefore, this study aims to establish an efficient protocol for the in vitro propagation of *Gymnocalycium* cv. Fancy plants with different levels of variegation are used. This will be achieved by evaluating the organogenic response and in vitro behavior of diverse explants (apical, central disc, hypocotyl, and epicotyl) from plants of different sizes (small, medium, and large) under three previously tested plant growth regulators (PGRs) used on chlorophyll-containing plants of the same cultivar [25]. This commercial hybrid developed by Cactusloft O.E. (Cullera, Valencia, Spain) originated from controlled crosses between *Gymnocalycium mihanovichii* and *Gymnocalycium fiedrichi* (Werdermann) Pažout, resulting in progenies with diverse morphologies and colorations due to their broad genetic background. This circumstance provides this cultivar with enormous potential from a commercial point of view, given that plants with different degrees of variegation can be obtained and selected, and color variants with different morphologies can also be identified [25] (Figure 2).

This information could be highly relevant in a commercial context, potentially improving graft propagation efficiency and optimizing the production of partially variegated plants for cactus collectors. Furthermore, the results could be applicable to other cactus species that may attract interest from consumers and collectors, addressing a significant gap in the current scientific literature on ornamental cacti micropropagation.

## 2. Results and Discussion

### 2.1. Explant Activation and Frequency of Response

Activated explants were assessed as the number of explants that resulted in some type of response (whether it be the formation of shoots, callus, or both), as well as their proportion relative to the total number of explants included for each factor and evaluated variable, are shown in Table 1. It is observed that the presence of TDZ1 in the medium significantly favored the activation of explants, while the other treatments showed a lower activation efficiency, but slightly higher than those observed in the control group (Table 1). The high activation capacity of TDZ compared to other growth regulators, both in generating callus or forming shoots, has been reported in various studies with other cactus species [6,8], although it was particularly prominent in previous trials using similar *Gymnocalycium* cv. Fancy plants [25]. Thus, similar results have been obtained in this study, consistent with previous findings.

In contrast, there was no difference in the activation capacity of explants obtained from medium-sized (69%) or large plants (70%), although the explants from hypocotyl and epicotyl derived from smaller plants showed a lower response (37%) (Table 1). Furthermore, apical and central disc explants obtained from longer plants also showed high rates of activation (64–74%) (Table 1). These findings suggest that hypocotyl and epicotyl explants, composed of younger and less mature tissues, are not efficient in activating their areoles with the use of cytokinins alone, although specific combinations of auxins and cytokinins may increase efficiency, as observed in other species [26,27,28]. In fact, some authors have reported responses using different auxin/cytokinin ratios [29,30,31,32,33,34].

Considering the variegation percentage of the starting plants, it was found that the absence of chlorophyll was extremely limiting. Thus, fully variegated explants responded worse (approximately 11%) to treatments compared to plants with chlorophyll tissues (with a response rate >60%), regardless of their color proportion (Table 1). This fact demonstrates that the total absence of chlorophyll hinders the normal development of plants on their own roots, due to the inability to carry out photosynthetic activity, which leads to their maintenance and propagation through grafting. This study has shown that fully variegated plants also exhibited very limited in vitro development, with many explants degenerating and others unable to grow beyond 8 mm. The inability to generate photo-assimilates might cause these already delicate plants to have a higher sensitivity to cuttings, as has been observed in *Agave angustifolia* Haw. albino variant somaclones [24].

#### 2.1.1. Calli Occurrence

The occurrence of calli was found to be closely related to the presence of TDZ1 in the culture medium (Figure 3a), as reported in previous works [35,36,37]. From the beginning of the induction period, calli formation in explants grown under TDZ1 was higher than that observed with other PGRs and in the control group (Figure 3a). Although activation in the presence of BAP8 was higher than that in the presence of KIN4 (with values very similar to those of the control group), the callus formation observed in these groups was minimal (Figure 3a). These results are in agreement with Giusti et al. [6], who reported a positive effect of TDZ on callus formation and shoot hyperhydration in *Escobaria minima* (Baird) D. Hunt, *Mammillaria pectinifera,* and *Pelecyphora aselliformis*, while the presence of BAP and KIN favored shoot formation.

During the development period in the absence of PGRs, the number of observed calluses tended to decrease in all treatments (except those exposed to KIN4), with a considerable reduction in the case of TDZ1 (Figure 3a). This effect may be related to the removal of hormones after the induction period, as callus formation is usually favored by the presence of cytokinins in the medium [8,35,38]. Through subsequent subcultures, the callus response gradually decreases, stimulating the structuring and organization of calluses into defined shoots, as observed in other works [25,35,36,37]. From the fourth month of the trial, no variations were detected in the number of calluses present on the explants.

The process of callogenesis was significantly higher in explants from medium- and large-sized plants than in epicotyl and hypocotyl explants (Figure 3b). These differences were mainly due to the higher response of apical explants compared to epicotyl explants, although, comparatively, central disc explants also showed better responses than hypocotyl explants (Figure 3d). Moreover, apices from medium-sized plants yielded better results than those from large plants. This fact is usual with cacti, where it has been observed that younger areoles are more sensitive to hormonal treatments than older ones [39]. In fact, most micropropagation protocols for *Opuntia* and *Hylocereus* are based on the use of young plants or cladodes [16,17,18,40].

The occurrence of callus based on the variegation percentage of the initial plants showed that the explants from the control group and the group with 25% variegation showed a higher average number of calluses at the end of the trial compared to the explants with a higher percentage of coloration (Figure 3c). Although statistically significant differences were not observed during the induction period, the development of the explants in the following months varied depending on the color percentage. In fact, plants with lower variegation (control group and 25% variegation) reduced the number of detectable calluses to a lesser extent (45%) than the other groups with higher degrees of variegation (50%, 75%, and 100%), resulting in a 94% reduction in callus presence. These results suggest that plants with a higher percentage of chlorophyll-containing tissue are more stable, and the possibility of reverting calluses to defined shoots after a callogenic process is lower. Therefore, calluses obtained from explants with a higher degree of variegation would exhibit greater reversibility to shoots, presumably due to their being more sensitive to cytokinins than calluses from chlorophyll-containing tissue. This fact was also observed by Rouinsard et al. [41], who reported different responses and behaviors when comparing the in vitro micropropagation of *Yucca gloriosa* ‘Variegata’, *Phormium tenax* ‘Jessie, and *Cordyline australis* ‘Pink Passion’, three cultivars with variegated foliage. In this case, they confirmed that variegation stability was genotype-dependent and highly related to the ability of the explants to be propagated by adventitious meristems [41].

#### 2.1.2. Shoot Production

The average monthly shoot production per explant for each evaluated factor is shown in Figure 4. The results demonstrate a significant contribution of hormonal effects on explant activation, both during the induction and development periods in the absence of regulators. In the first two months of cultivation under hormone conditions, explants treated with TDZ1 showed lower shoot production compared to other treatments and the control group. However, after subculturing in a medium without regulators, TDZ1-activated explants started responding positively and eventually showed the best average results at the end of the assay (Figure 4a). This increase in shoot number is closely linked to the observed reduction in the number of calli.

Therefore, the total shoot count at the end of the trial is not only due to new areola activation in explants but also to callus structuring and differentiation into shoots. Similar results were obtained in the evaluation of chlorophyllous plants [25] and *Rauvolfia serpentina* (L.) Benth. ex Kurz plants [35], where the effect of TDZ1 agrees without findings. In contrast, BAP8 and the control group resulted in very similar shoot production, while KIN4 showed slightly lower values (Figure 4a). Considering that responses to exogenous hormones can differ depending on the species and explant source [8,42,43] TDZ1 treatment would be the most efficient for propagating this type of plant material. The ability of TDZ to alter endogenous cytokinin metabolism [44,45,46,47,48] could explain its greater activation ability in explants compared to BAP8 or KIN4.

There were no statistically significant differences in the initial size of the plant material, although the average values obtained from small explants were slightly lower (Figure 4b). Therefore, even though the activation frequency of explants from small plants was noticeably lower than that from medium and large plants (Table 1), once activated, they enabled a productivity similar to that observed in explants from larger plants (Figure 4b).

When comparing the different types of explants, the highest productivity was observed in apical explants (2.80) from the start of the trial. The productivity of epicotyls (2.30) and central discs (2.26) was very similar, while hypocotyls had a slightly lower average (1.96) (Figure 4d). These results highlight the potential of epicotyl explants compared to apical and central disc explants, suggesting that they could be very interesting in commercial terms, with a more optimized protocol of activation. The ability to use small seedlings as a source material for micropropagation could considerably reduce the time between in vitro cultivation cycles, making the process faster and more profitable.

Regarding the influence of the color percentage of the starting material, no significant differences were observed between the various color groups and the control group (0% color) (Figure 4c). This situation demonstrates that the presence of variegation in plants does not limit their areolar response capacity. Even fully variegated explants were able to activate an average of three areolas per explant by the end of the trial (Figure 4c). Therefore, the limitation lies in the activation capacity of the explants (Table 1); once activated, they could potentially offer similar responses.

#### 2.1.3. Explant Efficiency

Since the number of areolas available on each type of explant is variable (Table 2), the ability to respond may also differ. From this perspective, we found that the efficiency of central discs was nearly doubled to that of apical and hypocotyl explants, while epicotyls showed a significantly lower frequency of areolar activation compared to the rest (Figure 5d). Considering the contribution of hormonal treatment and the type of explant, a graphical representation of productivity and efficiency results was conducted to show the interaction of these main factors in more detail (Figure 6).

The combination of both factors showed that in the absence of PGRs, explants generally exhibited lower shoot production compared to that observed after hormonal treatments. However, for apical explants, the control group showed the best results (Figure 6a). This fact in apical explants could be attributed to changes in the endogenous concentration of phytohormones that naturally activate this tissue after damage to the apical bud, which is common in cacti [36,37]. Therefore, this response is related to the intrinsic ability to activate dormant buds in that specific tissue fraction and not to the presence of a specific hormone in the medium.

In the evaluation of the average frequency of activated areolas, TDZ showed a greater ability to induce a response in all types of explants. Furthermore, a higher efficiency was observed in central disc explants compared to the rest of the explants (Figure 6b), even in those from the control group. The similarity in results between the central discs and apices from the control groups reinforces the hypothesis of the natural activation of dormant buds in plants with damaged apical meristems [36,37]. However, a slightly more efficient response was evident in the areolas from the central region of the plant (Figure 6b). These results are in agreement with previous trials conducted on chlorophyllous plants [25], although in this case, splitting the apices into two (thus eliminating apical dominance phenomena) allowed for a more comparable assessment of their efficiency compared to central discs.

Regarding the responses observed in explants from small plants, epicotyl explants showed a lower response compared to hypocotyls, both in terms of absolute production (Figure 6a) and frequency of areole activation (Figure 6b). These results could be linked to the higher callus production observed in epicotyls throughout the culture period (Figure 3d).

Finally, the specific combination of TDZ1 and central discs resulted in the best trial results, with percentages nearly double those obtained using other types of explants (Figure 5a and Figure 6b). Considering the significant activation capacity of central disc explants (Table 1), their high potential to activate their areolas in the presence of TDZ1 (Figure 6b), and thus their productivity in a strict sense (Figure 6a), the efficiency of this protocol would stand out above any other possible combination.

### 2.2. Root Emergence

Considering that the presence of roots could influence their response in generating shoots or calluses [49,50,51]. Root emergence was recorded during the first three months of in vitro cultivation, and some of the explants showed rhizogenesis. In general terms, the presence of TDZ1 in the medium had a highly detrimental effect on root formation, minimizing root emergence, while the absence of cytokinins promoted root development. In fact, the control group showed the highest values in the trial, followed by treatments with KIN4 and BAP8 (Figure 7a). TDZ is a synthetic hormone with both auxinic and cytokinin activity [52] that appears to have the ability to block or inhibit natural rooting mechanisms, as observed in previous studies [25]. From this perspective, the artificial origin of hormones could influence root emergence in the explants, as similar results have been obtained in other works with 2,4D [53]. However, the use of natural hormones like KIN or BAP seems to have a less negative impact, although they still provide lower rooting values compared to those observed in the control explants.

Conversely, it was found that the rooting capacity of the explants was closely related to the type of explant used, as apical and epicotyl explants showed a much higher response percentage (around 50–52%) compared to central disc (13%) and hypocotyl (22%) explants (Figure 7d). Rooting in epicotyl explants is expected, as root emergence is a natural process that spontaneously occurs in many species of cacti in response to damage or loss in the basal section of plants or their roots [54]. In contrast, the lower average rooting of hypocotyl explants could be due to their unnatural position on the medium. In contrast, rooting in apical explants was significantly higher than that observed in central discs, despite similar-sized explants placed in a natural position. This confirms that plant dissection often alters the proportion of endogenous hormones in these structures, favoring the establishment of high auxin/cytokinin ratios in the apices and consequently promoting root production in these types of explants [55,56]. On the contrary, this hormonal alteration does not occur in central discs, leading to a significant reduction in rooting events (Figure 7d).

Furthermore, rooting ability also appears to be related to the percentage of coloration of the explants used. In fact, explants from fully chlorophyllous plants rooted in a higher proportion than variegated plants (Figure 7c). Once again, these results seem to be associated with the greater vigor of chlorophyllous plants compared to plants with different degrees of variegation. Finally, when studying the rooting percentages in detail for each of the test groups, it could be determined that rooting events for *Gymnocalycium* occur more frequently in the absence of cytokinins and when using apices or epicotyls from fully chlorophyllous plants (Figure 8a–c). On the contrary, the presence of TDZ1 blocks root emergence and limits rhizogenesis.

### 2.3. Color Evaluation of the Obtained Shoots

A comprehensive evaluation of each individually activated areola was conducted to determine the productive capacity of variegated shoots, taking into account both the percentage of variegation in the starting plants and the tissue coloration observed in the areola itself. The total number of shoots exhibiting varying degrees of coloration, categorized based on the color of the original plant, is presented in Table 3. As expected, plants from the 0% and 100% groups were only able to activate G and C areolas, respectively. However, explants from partially variegated plants (25%, 50%, and 75%) had the ability to activate areolas from all three groups previously defined (G, M, or C), with the results included in Table 3. Therefore, it was essential to determine whether there was any correlation between the initial variegation percentage of the plants, the type of areola activated, and the degree of coloration of the obtained shoots.

Canonical correlation analyses revealed a strong correlation between the initial percentage of plant coloration and the type of activated areola, and between the initial percentage of plant coloration and the degree of coloration of the obtained shoots (Table 4). Furthermore, a strong correlation was also observed between the type of activated areola and the final coloration degree of the shoots (Table 4). Conversely, no correlation was detected between the applied hormonal treatment and the type of activated areola or the coloration degree of the obtained shoots (Table 4), indicating that hormonal treatment may not have a significant effect on obtaining shoots with different color gradients.

The table shows the estimated correlations between each set of canonical variables. *p*-values lower than *p* = 0.05 indicate a statistically significant correlation with a 95.0% confidence level.

In Figure 9, it can be observed that, as the initial plants exhibit a higher degree of variegation, the number of colored areolas activated also increases. It appears that there is no limitation on the activation of M or C areolas compared to G ones. Therefore, the activation process seems to be affected by factors unrelated to the color of the areola, and the activation of a higher number of colored areolas in plants with a higher percentage of initial variegation may be random. In this context, as the percentage of variegation in the initial plants increases, the likelihood of obtaining colored shoots also increases (Figure 10).

Since green areolas from the control group only resulted in chlorophyllous shoots and contributed to a remarkable percentage of the evaluated areolas (42%), an individualized evaluation of each type of independent areola and its corresponding generated shoot was conducted to avoid biases in the results, using only areolas from plants with some initial coloration (Table 5). The results showed that the coloration of the shoots was determined by the presence of chlorophyllous or variegated tissue in the areola (Figure 11). Therefore, green areolas (regardless of the variegation proportion of the initial plant) produced 97% green shoots, while colored areolas led to 98% colored plants, and from mixed areolas, 41% shoots were green and 59% shoots showed a range of degrees of coloration (Figure 11).

These results suggest the occurrence of somatic mosaicism in the areolar tissue. Therefore, at least two different cellular lines (chlorophyllous and variegated) would converge in the buds, capable of generating shoots with varying degrees of variegation. The proportion of these cell lines and their distribution in the bud would determine the greater or lesser variegation of the obtained shoots (Figure 12) [41,57,58]. These findings are highly interesting in a commercial context, as they could lead to the selection of the type of areola to activate based on the desired coloration of shoots, thereby optimizing large-scale production processes. Thus, the use of central discs would optimize the propagation of colored *Gymnocalycium*, in comparison to apices, where selecting areolas would be more difficult due to their smaller size and less inter-areolar space.

Additionally, shoots with new, different colorations and color patterns were obtained in this trial (Figure 13), demonstrating that it is possible to produce color variants from plants that appear to have similar variegation. This fact highlights the complexity of the underlying mechanisms involved in the structuring and development of new shoots in variegated plants. Therefore, having a deep understanding of how various cell lines related to the appearance of new shoots with different colorations or color patterns can be activated is essential for carrying out breeding programs aimed at obtaining new cultivars with different colorations.

Moreover, fully or partially variegated variants are of great commercial interest. In fact, many cultivars of *Gymnocalycium* with different colors (Hybotan, Seolhong, Damdan, Hwangweol, etc.) or the yellow peanut cactus (*Chamaecereus silvestrii* f. Lutea), have gained significant market relevance in recent decades [9]. According to the obtained results, the propagation processes of forms or cultivars already in circulation could be optimized through the selection of explants with fully colored areolas. Furthermore, this protocol could be adapted for other variegated cactus species with potential in the wholesale market.

However, there is a significant market associated with cacti for hobbyists, where customers are willing to pay higher prices for plants with distinctive and particular characteristics. Typically, variegation is among the aspects that collectors value. However, fully variegated plants (unable to survive on their own roots) usually do not appeal to this group, given the need for grafting to sustain them. The results obtained show that selecting mixed areolas as starting explants in a micropropagation protocol would allow for obtaining a very significant percentage of shoots with partial variegation (44% in this trial). These shoots can be isolated, rooted, and acclimatized, making their *ex vitro* development entirely viable. This would meet the expectations of collectors, opening up a new market niche associated with the colored cacti.

### 2.4. Acclimatization of the Obtained Shoots

Variegated shoots can be isolated, rooted, and acclimatized, making their ex vitro development viable (Figure 14). In fact, the acclimatization success rates were high: 19/20 for non-variegated shoots (0%), 17/20 for shoots with less than 50% variegation, and 18/20 for shoots with more than 50% variegation. These results demonstrate that the protocol can reliably produce a range of variegated plants with high survival rates during the critical acclimatization phase. Moreover, the developed methodology could potentially be extrapolated to other variegated cactus species of interest to collectors. The ability to consistently produce variegated plants on their own roots not only meets the expectations of collectors but also opens up a new market niche associated with colored cacti. This approach could significantly impact the ornamental plant industry by providing a reliable method for mass-producing variegated specimens, thereby meeting the growing demand for these cactus varieties in the market.

## 3. Materials and Methods

### 3.1. Plant Material and Disinfection

Plants were obtained by in vitro sowing of *Gymnocalycium* cv. Fancy seeds were kindly donated by Cactusloft OE (Cullera, Valencia, Spain). For disinfection, seeds were treated under aseptic conditions in a laminar flow cabinet (model AH-100, Telstar, Terrassa, Spain) for 1 min in 70% ethanol (*v*/*v*), followed by 25 min in 15% domestic bleach solution (*v*/*v*; 4% sodium hypochlorite) supplemented with 0.08% of the surfactant Tween-20 (*v*/*v*). Finally, seeds were rinsed 3 times in distilled sterilized water, with each rinse lasting 5 min, before sowing.

### 3.2. In Vitro Establishment and Culture Conditions

Murashige and Skoog (MS) basal media (Duchefa Biochemie, Haarlem, The Netherlands) [59] at half strength (1/2MS, 2.2 g L^−1^) supplemented with 15 g L^−1^ of sucrose (Sigma-Aldrich, St. Louis, MO, USA) and 7 g L^−1^ of bacteriological agar (Duchefa Biochemie, Haarlem, The Netherlands) was used as a sowing media. pH was adjusted to 5.7 before autoclaving at 120 °C for 20 min [60]. The disinfected seeds were sown in sterile plastic disposable Petri dishes, with 20 seeds in each dish. Seedlings developed under in vitro conditions inside a growth room at 26 ± 2 °C on shelves with a 16 h light (LED lamps)/8 h dark photoperiod and photosynthetic photon flux of 50 molm^−2^ s^−1^ for 8 months. Seedlings were subcultured monthly in fresh media.

### 3.3. Type of Explants

After an 8 months period of in vitro growth, plants were selected and classified depending on their initial size (Table 6). From medium and large-sized plants, two types of explants were obtained: apical and central disc. Each of these fragments was sectioned into two halves, resulting in 2 semicircular explants, i.e., four explants were obtained from each medium- or large-sized plant, with their size varying depending on the dimensions of the original plant (Table 6). From small-sized plants, epicotyl and hypocotyl were obtained by cutting the seedlings transversally into two parts, resulting in explants with diameters ranging from 4 to 8 mm (Table 6). The roots were completely removed in all cases. Therefore, four different types of explants were evaluated in this study: apical explants, central disc explants, epicotyls, and hypocotyls (Table 6). All explants were positioned to maximize the contact between the cut surface and culture medium.

### 3.4. Shoot Induction and Tissue Culture Conditions

To assess the morphogenic potential of variegated seedlings of *Gymnocalycium* cv. Fancy with different degrees of coloration, three specific concentrations of cytokinins (Duchefa Biochemie Company, RV Haarlem, The Netherlands) that generated responses in chlorophyllous plants in previous works [25] were studied: 6-Benzylaminopurine 8 µM (BAP8), Kinetin 4 µM (KIN4) and Thidiazuron 1 µM (TDZ1).

The explants were placed on a culture 1/2 MS media (2.2 g L^−1^), supplemented with sucrose (15 g L^−1^), agar (7 g L^−1^), and each of the three cytokinins (BAP8, KIN4, and TDZ1), with the pH adjusted to 5.7. These induction media were used to stimulate the activation of explants for shoot production. In addition, a control group of explants cultivated in the absence of PGRs was included for each plant size and type of explant. Furthermore, the explants were placed to maximize contact between the sectioned tissue and culture medium. The culture in the induction medium was maintained for two months. A subculture was performed after the first four weeks to ensure that the hormone concentration remained constant throughout the induction period. After the induction period, explants were subcultured to the initial basal 1/2 MS media at pH 5.7 in the absence of PGRs.

### 3.5. Experimental Design

A total of 180 plants were visually classified into four groups depending on the proportion of coloration observed, but plants fully covered by chlorophyll were also included (Figure 15). Considering their initial sizes, the plants were randomly distributed in each group before obtaining the explants (Table 6). Fully colored plants were only included in the small-sized group, as their lack of chlorophyll provoked sizes <8 mm in diameter. Furthermore, green plants (0% color) were evaluated as a different control groups: (a) on one hand, medium-sized plants subjected to the presence of PGRs and, (b) on the other hand, plants of all evaluated sizes (small, medium and large-sized) in absence of PGRs (Table 6).

A total of 574 explants from plants with different color percentages, including 214 of each apical explant and central disc explant and 73 of each epicotyl and hipocotyl explant, were cultivated during the experiment (Table 6). The explants were analyzed considering the degree of variegation and size of the original plant for each of the hormonal treatments and the control group. The explants were assigned to different treatments using a random number system to ensure a random distribution of samples among the experimental groups (Table 6). They were distributed in groups of four explants per Petri dish for their evaluation.

The frequency of activated explants was calculated as the number of explants that showed some type of response during the assay, either organogenic (shoots and roots appearance) or callogenic (callus tissue appearance), relative to the total number of explants (Table 1). The appearance of shoots and the formation of calluses were observed monthly for 5 months, only from those explants that responded to the treatments. Considering the different number of areolas present in each of the starting explants (Table 2), the results were interpreted taking into account both (i) productivity (i.e., total number of shoots per explant) and (ii) efficiency (i.e., ratio of the number of areoles that produced shoots with respect to the total number of areoles of each explant). The emission of roots was also recorded in terms of frequency for the first three months.

#### Areole Evaluation

The maximum response capacity of each explant type was determined by the number of areoles present on each (Table 2), as the dormant buds of cacti are contained within these structures. Therefore, in addition to evaluating productivity in terms of the number of calluses and shoots, the efficiency of the explants was analyzed by calculating the frequency of activated areoles (Figure 5). This frequency was calculated by comparing the number of areoles that gave rise to new shoots with the total number of areoles available on each explant. This measure provides a more precise view of the regenerative capacity of each explant type, taking into account its intrinsic potential based on the number of areoles present.

Using a magnifying glass (Kern OZO, 551), all activated areolas from colored plants were visually classified into three groups based on their coloration: green areolas “G” (where both the mamilla and areola were completely green), mixed areolas “M” (showing a combination of chlorophyllous and variegated tissue in both the mamilla and areola), and colored areolas “C” (where both the mamilla and areola were fully colored) (Figure 12). Chlorophyllous plants from the control groups were not included in this evaluation, since all the sprouts obtained in previous trials were completely green [25].

Subsequently, the obtained shoots were counted considering the percentage of coloration of the source plants and classified into four groups based on their final coloration: completely green shoots (without variegation, group “S0”), shoots with a coloration percentage below 50% (group “S1”), shoots with a coloration percentage above 50% (group “S2”), and completely colored shoots (without chlorophyllous tissue, group “S3”) (Figure 16). The relationships between the percentage of coloration of the activated areolas and the coloration of the shoots obtained based on the percentage of color of the initial plants, as well as the coloration of the shoots obtained based on the coloration of the activated areolas, were evaluated (Table 4).

### 3.6. Statistical Analysis

In order to analyze our data sets, multivariate ANOVA analysis was performed to check the effect of the different factors at a level of *p* < 0.05. The software used for performing this ANOVA analysis was Statgraphics Centurion XVIII (Statgraphics Technologies Inc., The Plains, VA, USA, https://www.statgraphics.com/download18, accessed on 22 January 2025). The presence of the three different hormones (BAP8, KIN4, and TDZ1) in the culture media, the relevance of the initial size (small, medium, and large-sized plants), the presence of diverse degrees of variegation in the initial plants, and the activation capacity of the various explants (apical explants, central disc explants, epicotyls, and hipocotyls) were analyzed only in those explants that responded to some treatment.

Means differing significantly were compared using the Student–Newman–Keuls test with a probability level of the 5%. Transformation of the data was previously performed to normalize the dataset using the following formulas:-For numerical and absolute data, including shoot emission, callus production, and averages.$$\sqrt{Y+\frac{1}{2}}$$-For frequency and efficiency values, including rooting capacity,
$$\sqrt[{arcsin}]{\frac{percentage}{100}}$$

The linear combinations between the initial color of the plants, the color of the activated areolas, the color of the shoots obtained, and the hormones used in the trial were studied through canonical correlation analysis, establishing a confidence level of 95%. Statgraphics Centurion XVIII (Statgraphics Technologies Inc., The Plains, VA, USA) was used for the analysis. To ensure the validity of our statistical analyses, we conducted Levene’s test to verify the homogeneity of the variances across treatments for each variable. Furthermore, to address potential non-homogeneity in plant size, direct comparisons between small and large plants were avoided. Instead, our analyses focused on color-related variables (initial plant coloration, color of activated areoles, and color of obtained shoots), which were consistently measurable across all experimental units, regardless of plant size.

### 3.7. Acclimatization

For acclimatization, 20 regenerated shoots from each of the established variegation percentages (0%, less than 50%, and more than 50%) were selected and isolated for further study. These shoots were previously rooted in full-strength MS medium supplemented with 15 g L^−^^1^ of sucrose and 7 g L^−^^1^ of bacteriological agar and pH adjusted to 5.7, for 4 weeks. Fully variegated shoots were excluded due to their inability to survive on their own roots.

The rooted shoots were first cleaned with sterile water before being transferred to cultivation trays containing a 1:1 mixture of peat and vermiculite. To regulate environmental moisture, the trays were placed in small-scale greenhouses. The acclimatization process spanned two weeks, during which the humidity was progressively lowered from an initial 100% to match ambient conditions by incrementally opening the greenhouse covers. Following this two-week period, the covers were completely removed, and the plants were allowed to continue their growth under standard growth chamber conditions. The plants were then maintained for an additional eight-week period to further monitor their development and acclimatization success rates for each variegation category. After 3 months of culture, once the plants were acclimatized and established, they were transplanted into pots using a substrate composed of 50% pumice, 40% peat, and 10% silica sand for their rustication process in the greenhouse.

## 4. Conclusions

In this work, a specific micropropagation protocol focused on obtaining shoots with varying degrees of variegation in *Gymnocalycium* cv. Fancy plants was successfully optimized. The results demonstrate that central disc explants cultured in medium containing TDZ1 produced the best results in terms of initial explant activation, shoot productivity, and efficiency of areolar activation per explant. Furthermore, it was observed that the coloration percentage of the starting plants (excluding completely achlorophyllous plants) did not limit the response capacity of the evaluated explants. Hence, a strong correlation exists between the initial variegation percentage of the plants and the type of activated areola.

Additionally, the type of activated areola correlated with the color percentage of the obtained shoots, highlighting a situation of somatic mosaicism in the areolas that determines the variegation percentage of the final shoots. These results provide a basis for adjusting and optimizing propagation protocols to obtain plants with different variegation proportions, based on specific commercial objectives.

Therefore, this protocol offers significant commercial value by enabling the control and prediction of plant coloration, which can be directed toward high-impact markets (wholesale or collector). Therefore, by selecting colored areolas, fully variegated shoots can be obtained, while explants carrying mixed areolas can produce shoots with partial variegation. Furthermore, the successful acclimatization results demonstrate the viability of ex vitro shoots, enhancing the protocol’s commercial applicability. Consequently, achieving greater efficiency and resource optimization in commercial plant production processes is possible.

## Figures and Tables

**Figure 1 plants-14-01091-f001:**
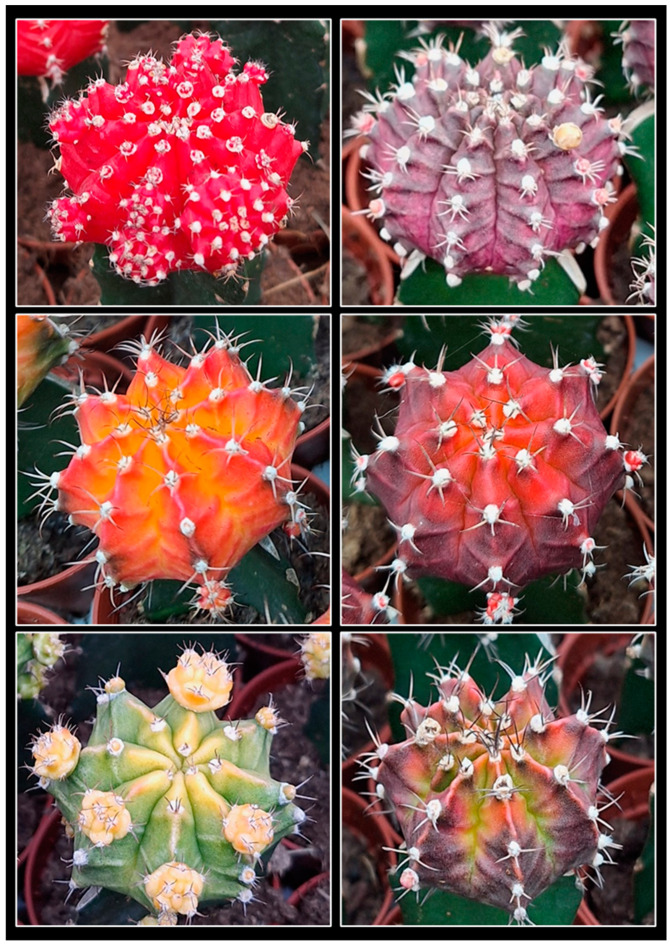
Commercial selections of *Gymnocalycium mihanovichii* with different colorations and color-distribution patterns.

**Figure 2 plants-14-01091-f002:**
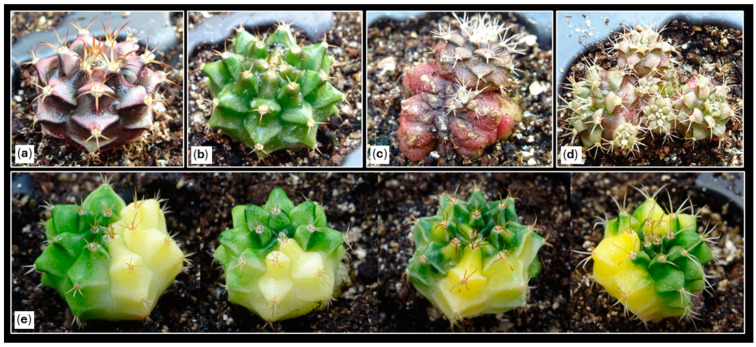
Morphological heterogeneity observed in one-year-old *Gymnocalycium* cv. Fancy shoots obtained in vitro and subsequently acclimatized. (**a**,**b**) Expected morphologies within the usual range of variation for the cultivar. (**c**,**d**) Unexpected morphologies obtained: monstrous (**c**) and caespitose forms (**d**). (**e**) Variegated plants with different degrees of variegation.

**Figure 3 plants-14-01091-f003:**
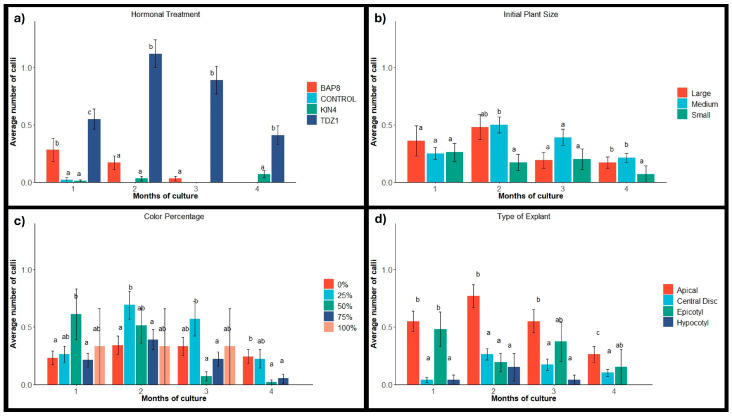
Average number of calli of *Gymnocalycium* cv. Fancy with their standard errors obtained monthly per factor and condition. (**a**) Hormonal treatments: BAP8 (6-Benzylaminopurine at 8 µM), KIN4 (Kinetin at 4 µM), and TDZ1 (Thidiazuron at 1 µM). Control: in the absence of PGRs; (**b**) Initial plant sizes: large (12–16 mm), medium (8–12 mm), and small (4–8 mm); (**c**) Initial plant variegation (color percentage): 0%, no variegation; 25%, plants with more chlorophyll than non-chlorophyll tissue; 50%, plants with equal proportion of chlorophyll and non-chlorophyll tissue; 75%, plants with more than 50% non-chlorophyll tissue; 100%, plants completely variegated; (**d**) Type of explant evaluated. Values within each month of culture, followed by different letters, are statistically different at *p* = 0.05 according to the Student–Newman–Keuls test.

**Figure 4 plants-14-01091-f004:**
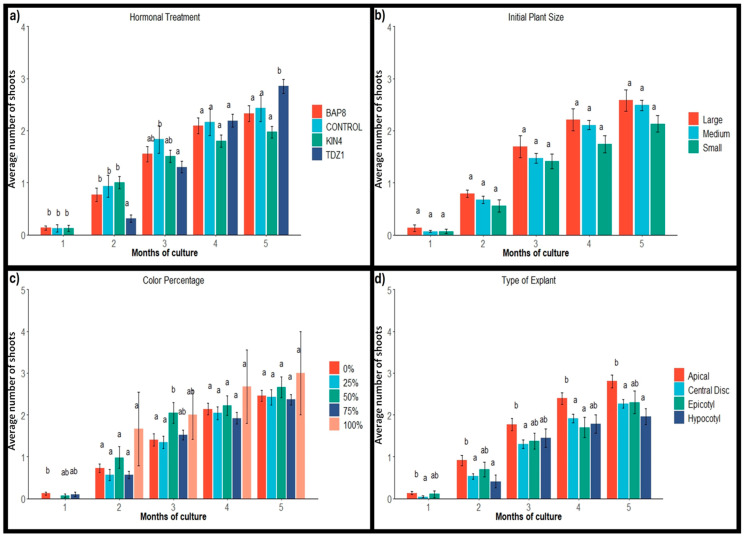
Average number of shoots of *Gymnocalycium* cv. Fancy with their standard errors obtained monthly per factor and condition. (**a**) Hormonal treatments: BAP8 (6-Benzylaminopurine at 8 µM), KIN4 (Kinetin at 4 µM), and TDZ1 (Thidiazuron at 1 µM). Control: in absence of PGRs; (**b**) Initial plant sizes: Large (12–16 mm); Medium (8–12 mm); Small (4–8 mm); (**c**) Initial plant variegation (Color percentage): 0%, no variegation; 25%, plants with more chlorophyll than non-chlorophyll tissue; 50%, plants with equal proportion of chlorophyll and non-chlorophyll tissue; 75%, plants with more than 50% non-chlorophyll tissue; 100%, plants completely variegated; (**d**) Type of explant evaluated. Values within each month of culture, followed by different letters, are statistically different at *p* = 0.05 according to the Student–Newman–Keuls test.

**Figure 5 plants-14-01091-f005:**
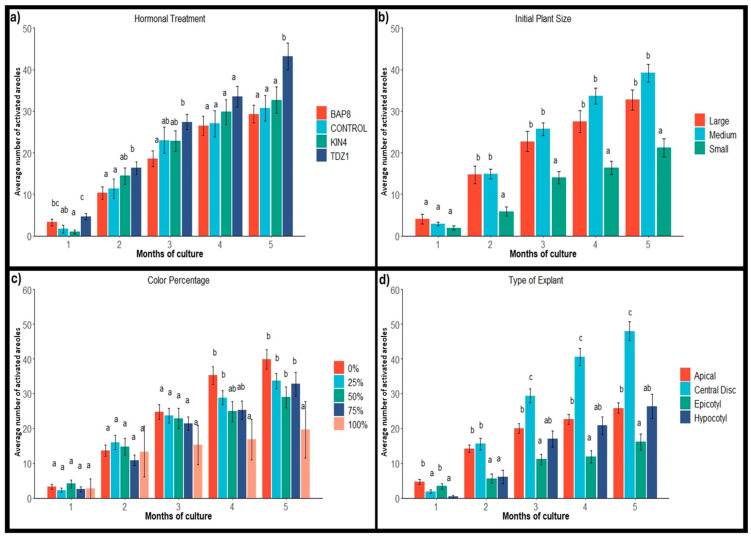
Average number of the cumulative frequency of activated areoles (areoles that gave rise to new shoots with respect to the total number of areoles available on each explant) of *Gymnocalycium* cv. Fancy with their standard errors obtained monthly by factor and condition. (**a**) Hormonal treatments: BAP8 (6-Benzylaminopurine at 8 µM), KIN4 (Kinetin at 4 µM), and TDZ1 (Thidiazuron at 1 µM). Control: in absence of PGRs; (**b**) Initial plant sizes: Large (12–16 mm); Medium (8–12 mm); Small (4–8 mm); (**c**) Initial plant variegation (Color percentage): 0%, no variegation; 25%, plants with more chlorophyll than non-chlorophyll tissue; 50%, plants with equal proportion of chlorophyll and non-chlorophyll tissue; 75%, plants with more than 50% non-chlorophyll tissue; 100%, plants completely variegated; (**d**) Type of explant evaluated. Values within each month of culture followed by different letters are statistically different at *p* = 0.05 according to the Student–Newman–Keuls test.

**Figure 6 plants-14-01091-f006:**
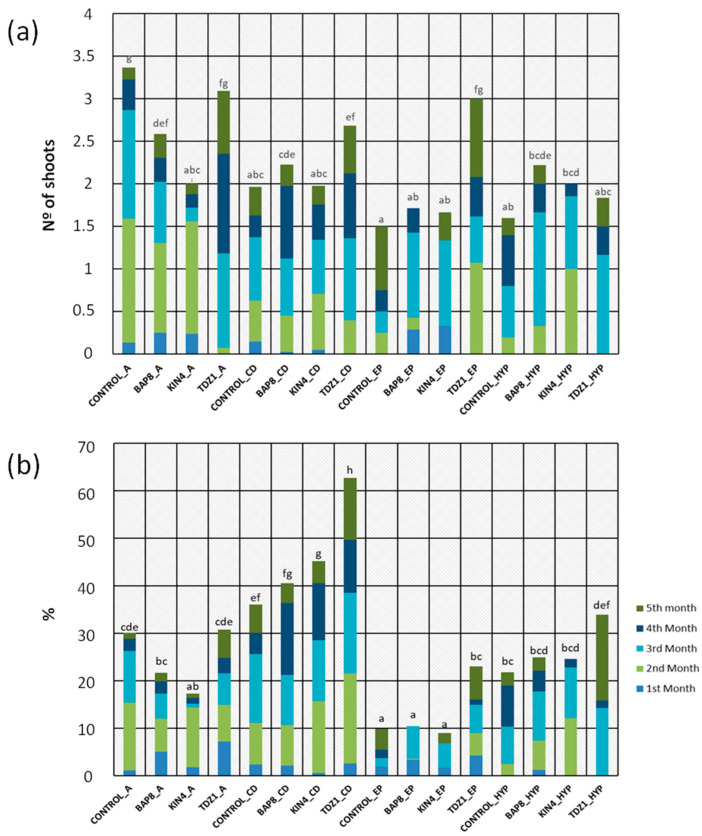
(**a**) Average number of shoots of *Gymnocalycium* cv. Fancy obtained for each “Hormone + Explant Type” combination during each month of cultivation. (**b**) Average frequency of areoles activated by each “Hormone + Explant Type” combination during each month of cultivation**.** Hormones: CONTROL (control group), BAP8 (6-Benzylaminopurine), KIN4 (Kinetin), and TDZ1 (Thidiazuron). Numbers following the conditions indicate the hormone concentration (1, 4, or 8 µM). Capital letters indicate the explants used in each combination: A (apical), CD (central discs), EP (epicotyl), and HYP (hypocotyl). Letters (a, b, c, d, e, f) above the bars represent significant differences (*p* = 0.05) based on sample means at the fifth month of evaluation, according to the Student–Newman–Keuls test.

**Figure 7 plants-14-01091-f007:**
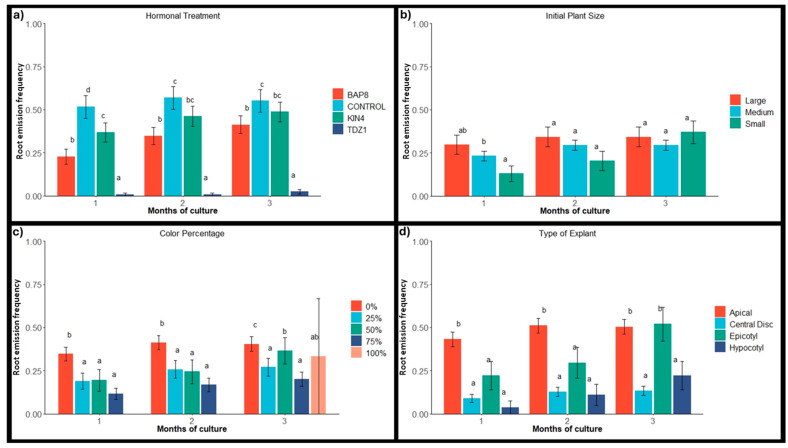
Average and standard error of root emission frequency (calculated as the number of rooted explants based on the total number of explants) obtained monthly by factor and condition in *Gymnocalycium* cv. Fancy. (**a**) Hormonal treatments: BAP8 (6-Benzylaminopurine at 8 µM), KIN4 (Kinetin at 4 µM), and TDZ1 (Thidiazuron at 1 µM); (**b**) Initial plant sizes: Large (12–16 mm); Medium (8–12 mm); Small (4–8 mm); (**c**) Initial plant variegation (Color percentage): 0%, no variegation; 25%, plants with more chlorophyll than non-chlorophyll tissue; 50%, plants with equal proportion of chlorophyll and non-chlorophyll tissue; 75%, plants with more than 50% non-chlorophyll tissue; 100%, plants completely variegated; (**d**) Type of explant evaluated. Values within each month of culture, followed by different letters, are statistically different at *p* = 0.05 according to the Student–Newman–Keuls.

**Figure 8 plants-14-01091-f008:**
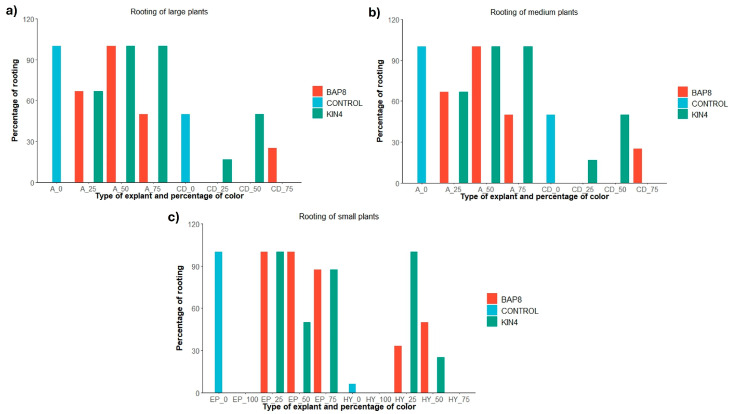
Percentage of rooting in *Gymnocalycium* cv. Fancy at the third month of culture for each combination of factors. The letters refer to the type of explant: A (apices), CD (central discs), EP (epicotyl), and HY (hypocotyl). (**a**) Rooting of explants from large plants (12–16 mm); (**b**) Rooting of explants from medium plants (8–12 mm); (**c**) Rooting of explants from small plants (4–8 mm). The numbers accompanying the letters indicate the initial percentage of variegation in the plants: 0 (no variegation), 25 (plants with more chlorophyll than non-chlorophyll tissue), 50 (plants with equal proportions of chlorophyll and non-chlorophyll tissue), 75 (plants with more than 50% non-chlorophyll tissue), and 100 (plants completely variegated). Hormonal treatments: BAP8 (6-Benzylaminopurine at 8 µM) and KIN4 (Kinetin at 4 µM). CONTROL: Control group without variegation.

**Figure 9 plants-14-01091-f009:**
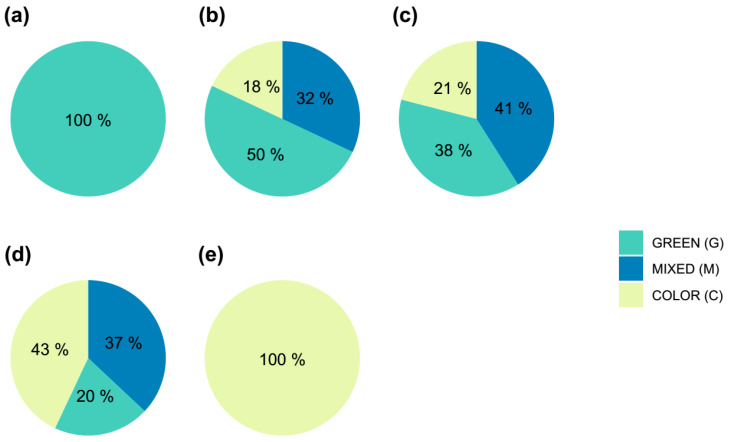
Frequency of activated areoles of each type (green = G, mixed = M, or color = C) in *Gymnocalycium* cv. Fancy as a function of the initial coloration of the plants. (**a**) Green plants without variegation; (**b**) plants with 25% of variegation (**c**) plants with 50% of variegation, (**d**) plants with 75% of variegation, and (**e**) plants completely colored (100% variegation).

**Figure 10 plants-14-01091-f010:**
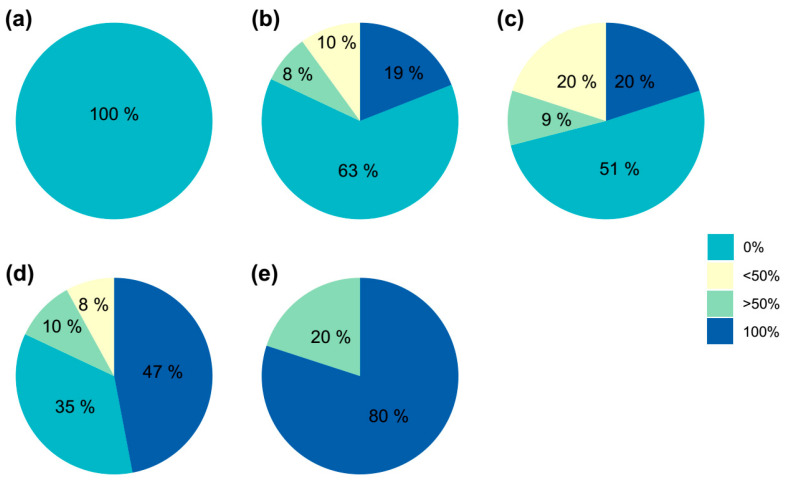
Percentage of shoots obtained with different degrees of variegation (0%, less than 50%, more than 50%, and totally variegated, 100%) in *Gymnocalycium* cv. Fancy with respect to the percentage of variegation of the initial plants. (**a**) Green plants without variegation; (**b**) plants with 25% variegation (**c**) plants with 50% variegation, (**d**) plants with 75% variegation, and (**e**) plants completely colored (100% variegation).

**Figure 11 plants-14-01091-f011:**
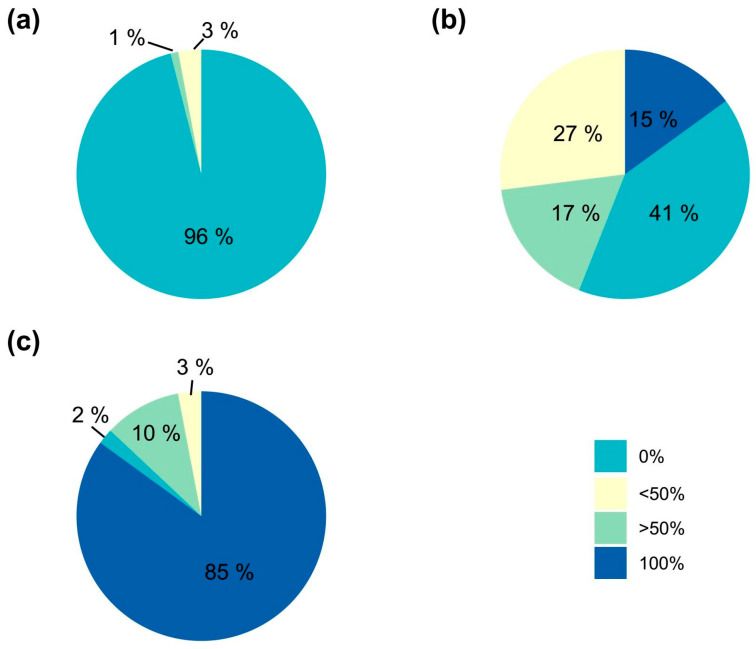
Percentage of shoots obtained with different degrees of variegation (0%, less than 50%, more than 50%, and totally variegated, 100%) in *Gymnocalycium* cv. Fancy with respect to the coloration of the starting areoles ((**a**)—Green; (**b**)—Mixed; (**c**)—Color).

**Figure 12 plants-14-01091-f012:**
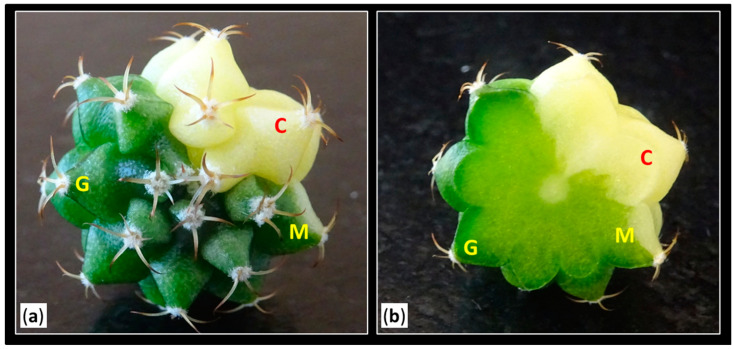
Classification of the types of areolas of *Gymnocalycium* cv. Fancy according to their coloration. (**a**) View of the apical part of the plant. (**b**) Transverse section of a variegated plant. Letters indicate: “G”, green areolas (where both the mamilla and areola were completely green); “M”, mixed areolas (showing a combination of chlorophyllous and variegated tissue in both the mamilla and areola); and “C”, colored areolas (where both the mamilla and areola were fully colored).

**Figure 13 plants-14-01091-f013:**
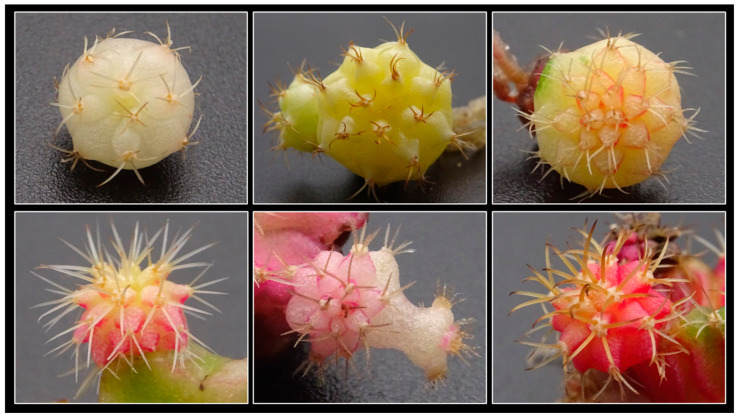
Varied shoots of *Gymnocalycium* cv. Fancy with different colorations and different color patterns observed in the trial.

**Figure 14 plants-14-01091-f014:**
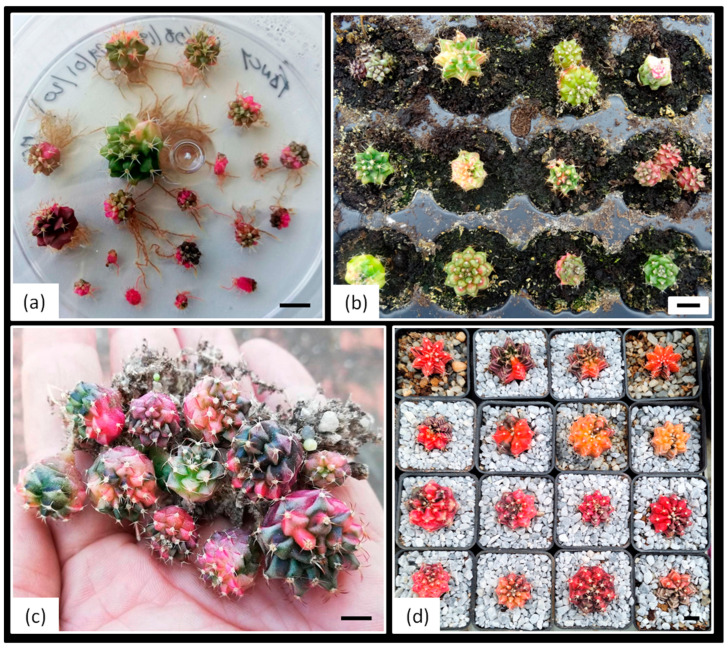
In vitro rooting and ex vitro acclimatization process of variegated explants of *Gymnocalycium* cv. Fancy. (**a**) Rooted shoots on MS medium after 4 weeks of in vitro culture; (**b**) Development of rooted explants in plug trays after one month of *ex vitro* culture; (**c**) Established plants after 3 months of culture; (**d**) Fully acclimatized plants growing in greenhouse conditions at 6 months. Scale bars = 10 mm.

**Figure 15 plants-14-01091-f015:**
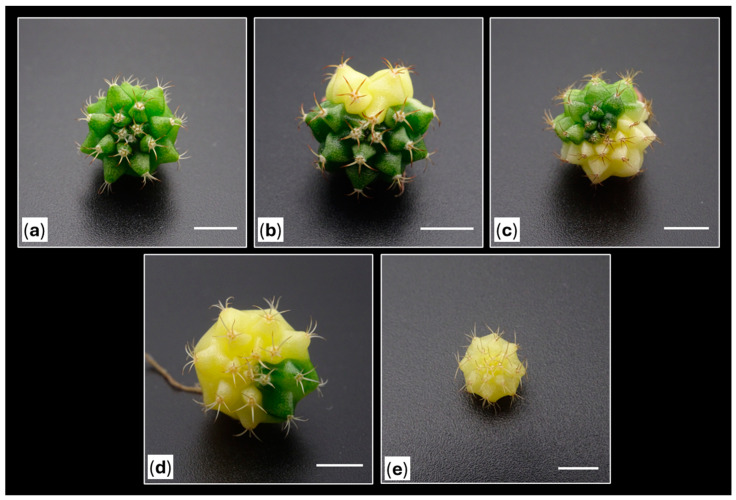
Initial plant variegation percentage classification system in *Gymnocalycium* cv. Fancy: (**a**) completely chlorophyll plant (no variegation, 0%), (**b**) plants with more chlorophyll than non-chlorophyll tissue (25% variegation), (**c**) plants with equal proportion of chlorophyll and non-chlorophyll tissue (50% variegation), (**d**) plants with more than 50% non-chlorophyll tissue (75% variegation), and (**e**) plants with completely non-chlorophyll tissue (100% variegation). Scale bars = 10 mm.

**Figure 16 plants-14-01091-f016:**
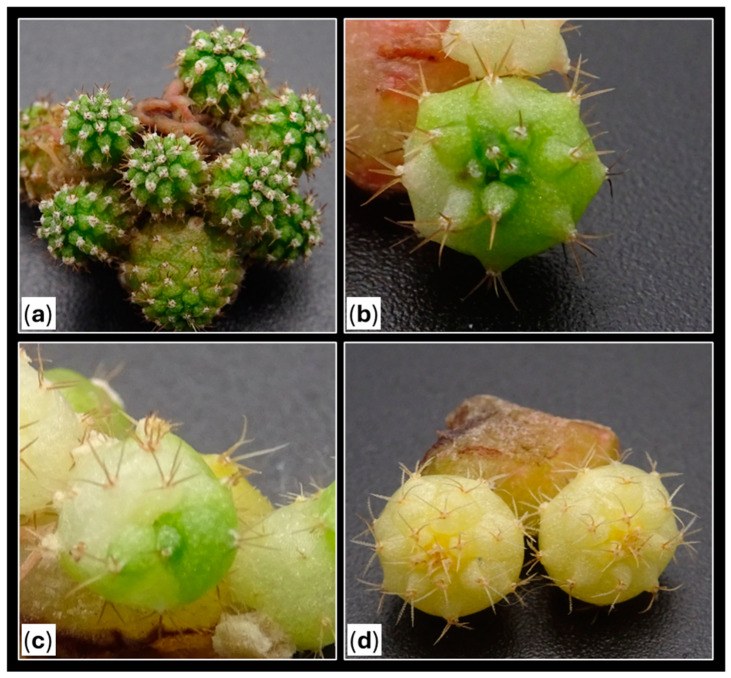
Grading system of the obtained shoots according to their final coloration in *Gymnocalycium* cv. Fancy. (**a**) completely green shoots (without variegation, group “S0”); (**b**) shoots with a percentage of coloration lower than 50% (group “S1”); (**c**) shoots with a percentage of coloration higher than 50% (group “S2”), and (**d**) completely colored shoots (without chlorophyll tissue, group “S3”).

**Table 1 plants-14-01091-t001:** Frequency of activated explants of *Gymnocalycium* cv. Fancy as a function of evaluated factors at the 5th month.

Factor	Total Number of Explants	Activated Explants	% of Response ^(1)^
Treatment ^(2)^			
BAP8	152	92	60.53 a
KIN4	150	76	50.67 a
TDZ1	148	123	83.11 b
CONTROL	124	58	46.77 a
Plant Size			
Large	96	67	69.79 b
Medium	332	228	68.67 b
Small	146	54	36.99 a
% of Color			
0	220	136	61.82 b
25	110	74	67.27 b
50	60	41	68.33 b
75	156	95	60.90 b
100	28	3	10.71 a
Type of Explant			
Apical	214	137	64.02 b
Central Disc	214	158	73.83 c
Epicotyl	73	27	36.99 a
Hypocotyl	73	27	36.99 a
Total	574	349	60.8

^(1)^ Values followed by the same letter are not statistically different for *p* = 0.05 according to the Student–Newman–Keuls; ^(2)^ BAP8 = 6-Benzylaminopurine at 8 µM; KIN4 = Kinetin at 4 µM; TDZ1 = Thidiazuron at 1 µM; CONTROL = Control explants grown in absence of PGRs.

**Table 2 plants-14-01091-t002:** Average number of areoles per explant of *Gymnocalycium* cv. Fancy according to the initial plant size and explant type.

Factor	Cases	Average ^(1)^
**Plant Size**		
Large	96	9.30 ± 0.43 a
Medium	332	9.37 ± 0.27 a
Small	146	10.21 ± 0.54 a
**Type of Explant**		
Apical	214	13.15 ± 0.25 b
Central Disc	214	5.51 ± 0.13 a
Epicotyl	73	15.30 ± 0.51 b
Hypocotyl	73	5.12 ± 0.43 a

^(1)^ Values followed by the same letter are not statistically different for *p* = 0.05 according to the Student–Newman–Keuls.

**Table 3 plants-14-01091-t003:** Number and type of activated areolas and percentage of coloration of obtained shoots in *Gymnocalycium* cv. Fancy in relation to the percentage of coloration of the initial plants.

% Color of the Original Plant	No. of Activated Areolas	Color of Activated Areola ^(1)^			Percentage of Shoot Coloration			
		C	M	G	0%	<50%	>50%	100%
0	368	0	0	368	368	0	0	0
25	177	31	57	89	112	18	14	33
50	107	22	44	41	55	21	10	21
75	217	94	79	44	77	18	21	101
100	5	3	2	0	0	2	1	2
Total	874	150	182	542	612	59	46	157

^(1)^ Color of activated areola: “G”, green areolas; “M”, mixed areolas; and “C”, colored areolas.

**Table 4 plants-14-01091-t004:** Canonical correlation between evaluated factors and their corresponding significance values (*p*-value) for each combination in *Gymnocalycium* cv. Fancy.

Interactions	*p*-Value
Color percentage of the initial plant × Color of the activated areolas	0.000
Color percentage of the initial plant × Color percentage of the obtained shoots	0.000
Color of the activated areolas × Color percentage of the obtained shoots	0.000
Hormonal treatment × Color of the activated areolas	0.133
Hormonal treatment × Color percentage of the obtained shoots	0.766

**Table 5 plants-14-01091-t005:** Number of shoots obtained from the initial variegated plants of *Gymnocalycium* cv. Fancy as a function of the type of activated areola. Number of shoots obtained from areoles derived from the initial variegated plants classified according to the type of areola (shoots from the activated areoles of the initial chlorophyllic plants are not included).

Type of Areola	Coloration of Shoots ^(1)^			
	S0	S1	S2	S3
Color (C)	3	4	15	129
Mixed (M)	74	48	31	28
Green (G)	166	5	1	0
Total	243	57	47	157
Total of shoots	504			

^(1)^ Coloring percentage of shoots: group S0 = green shots without variegation; group S1 = shoots with a color percentage below 50%; group S2 = shoots with a color percentage above 50%; and group S3 = completely colored, without chlorophyllous tissue.

**Table 6 plants-14-01091-t006:** Experimental design of the trial.

Plant Size	% Color	No. Plants	No. of Explants Evaluated ^(1)^				Treatment ^(2)^			
			A	CD	EP	HYP	BAP8	KIN4	TDZ1	CONTROL
Large (12–16 mm)	0	7	14	14	-	-	-	-	-	28
	25	9	18	18	-	-	12	12	12	-
	50	2	4	4	-	-	4	4	-	-
	75	6	12	12	-	-	8	8	8	-
Medium (8–12 mm)	0	40	80	80	-	-	32	32	32	64
	25	15	30	30	-	-	20	20	20	-
	50	7	14	14	-	-	8	8	12	-
	75	21	42	42	-	-	28	28	28	-
Small (4–8 mm)	0	16	-	-	16	16	-	-	-	32
	25	7	-	-	7	7	6	4	4	-
	50	12	-	-	12	12	8	8	8	-
	75	24	-	-	24	24	16	16	16	-
	100	14	-	-	14	14	10	10	8	-
Total for condition			214	214	73	73	152	150	148	124
Total trial		180	574				574			

^(1)^ Type of explant: A (apical), CD (central disc), EP (epicotyl), HYP (hypocoty). ^(2)^ Treatments: BAP8 (6-Benzylaminopurine at 8 µM), KIN4 (Kinetin at 4 µM) and TDZ1 (Thidiazuron at 1 µM).

## Data Availability

Data are contained within the article.

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
