# Peer review of "Response to In Vitro Micropropagation of Plants with Different Degrees of Variegation of the Commercial Gymnocalycium cv. Fancy (Cactaceae)"

_plants, 2025, doi:10.3390/plants14071091_

Round 1
Reviewer 1 Report (Previous Reviewer 1)
Comments and Suggestions for Authors
The revised version of the paper “Response to in vitro micropropagation of plants with different degrees of variegation of the commercial Gymnocalycium cv. Fancy (Cactaceae)” by Carles Cortés-Olmos, Vladimir Marín Guerra-Sandoval, Carla Guijarro-Real, Benito Pineda, Ana Fita, and Adrián Rodríguez-Burruezo, was considerable improved regarding the original one. The list of the point-to-point responses to the comments made to the original manuscript shows that the authors attended each comment/suggestion properly, which resulted in a much better written document. However, although the authors claim that the objective of establish a protocol for in vitro propagation protocol of Gymnocalycium plants was achieved, they do not show the experimental evidence to demonstrate such asseveration. In the new added section “2.4. Acclimatization of the obtained shoots” the authors stated “variegated shoots can be isolated, rooted, and acclimatized, making their ex vitro development entirely viable” but no results showing regenerated plants were shown.
Author Response
Dear Reviewer,
We sincerely appreciate your comment about the lack of experimental evidence to demonstrate the success of the in vitro propagation protocol. We acknowledge that this is a crucial aspect of our study and agree that it needed to be addressed more comprehensively.
To address this, we have included a new detailed figure that illustrates the complete propagation process, from the isolation of explants to their development in the greenhouse. This visual representation provides clear evidence of the successful stages of micropropagation and acclimatization.
Furthermore, we have expanded the "2.4. Acclimatization of the obtained shoots" section in Materials and Methods to offer a more comprehensive description of the acclimatization and rustication process.
We believe these additions provide the necessary evidence to support our claim about the success of the in vitro propagation protocol for Gymnocalycium cv. Fancy.
Thank you once again for your feedback, which has helped us improve the quality of our research manuscript.
Reviewer 2 Report (Previous Reviewer 2)
Comments and Suggestions for Authors
After the changes to the previous versions, I agree that the manuscript should be published. I just don't quite understand why the previous process was terminated and a new one created in the MDPI system; once this was done, it gave the impression that an adequate opinion had not been issued previously.
Author Response
Dear Reviewer,
We sincerely thank you for your time and effort in reviewing our manuscript throughout the revision process. Your feedback and suggestions have significantly contributed to improving the quality of our work.
We appreciate your approval for publication of this final version of the manuscript. Regarding your question about the termination of the previous process and creation of a new one in the MDPI system, this was due to administrative reasons beyond our control. We apologize for any confusion this may have caused and assure you that all previous reviews and revisions were taken into account in this final version.
Once again, we are grateful for your thorough evaluation and positive recommendation for publication.
Reviewer 3 Report (Previous Reviewer 3)
Comments and Suggestions for Authors
The manuscript reports on the micropropagation of variegated ornamental plants of Gymnocalycium cv. Fancy with the aim of establishing protocols for in vitro propagation of plants having different amounts of colour variegation. By carefully investigating different starting explants using specific plant growth regulators, significant percentages of variegated shoots were successfully obtained and the protocol optimized.
The present manuscript reporting sound conducted experimental work, is quite well written with some amelioration compared to the first submission.
Table headings and figure legends (3-11) should be improved by indicating the plant species.
Author Response
Dear Reviewer,
We sincerely thank you for your time and effort in reviewing our manuscript throughout the revision process. Your feedback and suggestions have significantly contributed to improving the quality of our work.
Regarding your suggestion about table headings and figure legends, we have addressed this by including "Gymnocalycium cv. Fancy" in each table and figure (3-11) to enhance clarity and comprehension. This addition will ensure that readers can easily identify the species throughout the manuscript.
Once again, we are grateful for your time and expertise throughout the review process. Your constructive feedback has significantly contributed to the enhancement of our manuscript.
Round 2
Reviewer 1 Report (Previous Reviewer 1)
Comments and Suggestions for Authors
In the previous revised versions of the paper “Response to in vitro micropropagation of plants with different degrees of variegation of the commercial Gymnocalycium cv. Fancy (Cactaceae)” the list of the point-to-point responses to the comments made to the original manuscript showed that the authors attended each comment/suggestion properly, which resulted in a much better document. However, experimental evidence showing results to achieve the main objective was not included in the MS.
In this third revised version, the authors finally included the experimental evidence to demonstrate the in vitro propagation of Gymnocalycium plants at the end of the results section, which points out that the objective of establishing a protocol for in vitro propagation of Gymnocalycium plants, now more extensively delineated at the end of the materials and methods section of this third revised version, was successfully achieved.
As a consequence, I consider this revised version of the manuscript ready to be accepted and may continue with the publication process.
Author Response
Dear Reviewer,
We sincerely appreciate your thorough review and valuable comments, which have been instrumental in enhancing our manuscript. We are pleased to hear that the inclusion of the requested experimental evidence now meets the requirements for acceptance.
Thank you again for your time and consideration.
This manuscript is a resubmission of an earlier submission. The following is a list of the peer review reports and author responses from that submission.
Round 1
Reviewer 1 Report
Comments and Suggestions for Authors
The paper “Response to in vitro micropropagation of plants with different degrees of variegation of the commercial Gymnocalycium cv. Fancy (Cactaceae)” by Carles Cortés-Olmos, Vladimir Marín Guerra-Sandoval, Carla Guijarro-Real, Benito Pineda, Ana Fita, and Adrián Rodríguez-Burruezo, reports the effect of the plant size and color, the explant type, and the cytokinin type over the organogenesis and the variegation of shoots in the in vitro cultured explants.
The objective was not clearly specified in the abstract section, where it was mentioned that the response of the variants of variegated plants has not been investigated in in vitro cultures. However, at the end of the introduction the authors indicated that the objective was to establish an efficient protocol for in vitro propagation of Gymnocalycium cv. Fancy plants with varying degrees of variegation through the evaluation of the effect of type of cytokinin and type of explants from plants with different sizes and variegation proportion, over the organogenic response of the in vitro cultivated explants. Authors also point out that “the aim is to determine the effect of these plant growth regulators on variegated plants and the relationship between the initial plants' variegation proportion and the productivity of shoots with varying degrees of variegation”
Since experimental evidences covering such objectives were reported by the authors in a previous report (A. Micropropagation and acclimatization of Gymnocalycium cv. Fancy (Cactaceae): Developmental responses to different explant types and hormone conditions. Plants 2023, 12, 3932), and the difference with the current study was not presented in the introduction section, and was lightly compared with the results in the manuscript; in addition that due to the ornamental value of the Gymnocalycium gender along with other plant species from the Cactaceae family, there are plenty of reports regarding to the in vitro propagation of these plant species, including the use of different type of explants and use of exogenous cytokinin regulators for activation the development of explants and dormant meristems in the areoles (see for example Micropropagation of cacti—a review. Haseltonia 2014.19 (2014): 46-63, and Method for vegetative propagation of plants. U.S. Patent Application 18/001,732 filed July 27, 2023), the present study lack of originality, and there are not a real novel/new technologic advances in this work.
Finally, although the manuscript shows a good understanding of the technical terms, writing and use of statistical analysis, the principal objective mentioned in the introduction section of the manuscript, which was to establish an efficient protocol for in vitro propagation of Gymnocalycium cv. Fancy plants with varying degrees of variegation, was not achieved. As pointed out at the end of the results section, with the results it could be possible to select mixed areolas as starting explants or shoots could be isolated, rooted, and acclimatized, making their ex vitro development to settle down a micropropagation protocol; however such protocol is still pending.
Therefore, I have to conclude that this manuscript cannot be published under the current status.
General comments
Some crucial background information is missing in the introduction section. For example, discussing studies about the diversity in morphologies and colorations of the plant species studied.
In the results section, consider using the word frequency instead of the word percentage.
Improve subtitles by giving more accurate context of the respective contents.
Define early in the text how the frequency or number of activated explants and activated areolas was established. For example, at the beginning of section 2.1. it should be starting by pointing out how the activation of explants were assessed (It was mentioned until 3.4. Experimental design section, row 538)
There are too many confusing and inaccurate sentences, especially in the material and methods section, where also several experimental details are missing.
Short explanation of the respective experiment and how the parameter shown were measured showed must be included in the Tables and Figures captions.
Because the experimental design used (Table 11), interaction between factors cannot be evaluated.
Table 5 show the results from the major effect of each factor over the frequency of activated areoles.
The results analyzing the combination of factors (R265-299) and the canonical correlation analysis (R361-428) deserves further explanation along the manuscript. Because the experimental treatments were performed in blocks it is difficult to combine the levels of independent factors such the plant sizes with all the explants type. Also, for canonical correlation analysis the measured variables should come from homogeneous experimental units. So, this condition should be demonstrated for the analyzed treatments. See additional comments in the text.
Notice that the last sentence in the results section point out that further research is needed to achieve the principal objective settle down at the beginning of this manuscript.
Some specific comments
Results
R102 Activated explants were assessed as the number....
R139 It should be noticed that the terms production and presence or occurrence are used for description of different parameters. In plant cell and tissue culture, production is used for biomass amount.
R240 Efficiency of what? Specify. Also, explain here, and in section 3.4.1, and captions of Tables 8 and 9 how areolar activation was assessed.
R242 … the theoretical response may also differ… The idea is confusing, please explain what theoretical response means.
R244 Explain the criteria used for assessing areolar activation.
R245 The idea is confusing. Which results were not maintained? Explain.
R247 Do you mean explants from medium and large sized plants? Explants with different sizes were not described in the material and methods section.
R259 Table 5. Is this the cumulative frequency of activated areoles? Explain in the caption the criteria used for assessing areolar activation.
R265- This section deserves further analysis and explanation. Experimental treatments were performed in blocks. Therefore, it is difficult to combine the levels of independent factors such the plant sizes with all the explants type. In fact, explants from small plants are different from the other plant sizes. Usually, combination of factors are used to estimate effects by interaction of such factors, which can not be done in the experimental design used in this work.
R361 Information about the canonical correlation analysis is needed. Usually, for this statistical procedure the measured variables should come from homogeneous experimental units. So, this condition should be demonstrated for the analyzed treatments.
Material and methods section
R471-474 This sentence should not be in the materials and methods section, and more information about this topic could be provided in the introduction section.
R489 Induction of what? Please, specify.
R491 How was the degree of coloration assessed? Please, cite Figure 9.
R495 What explants? How the explants to establish the tissue cultures were obtained should be explained here citing Table 11.
R497 Induction of what? Of the explants? Please explain. Are these this the induction media? Specify.
R497-498 Consider exchange by: Besides, a control group of explants was cultivated in the absence of PGRs for each plant size.
R508 Change for by from
R505-512 Consider moving this paragraph to the previous section, perhaps under a subtitle like type of explants
R513 How was the proportion of coloration settled down? Please stated that 0 color refers to plants fully covered by chlorophyll
R514 Why was no variegation (0% color) not included in the parenthesis? Please, to refer to the proportion of coloration consider exchanging the numbers in parenthesis by (Figure 9).
R518 ….included as a different control groups…
R518-520 Please, note and emphasize that treatment with medium sized plants in absence of PGRs is not present in Table 11. Explain why.
R521 Table 1. This is a quasi-experimental design, where explants were grouped based on non-random criteria after their plant position, plant size and vegetated proportion, therefore the statistical analysis must be performed by independent groups.
R532 …five 574 explants… Please revise the text. …five 574… is confusing.
R534 Explants were evaluated or used to evaluate something? Explain what characteristic or response variable was evaluated.
R539 consider specify the response variable measured, for example: ..organogenic (shoots and roots appearance) or callogenic (callus tissue appearance)…
R560-562. Point out were the results of the relationship are shown.
See additional comments in the attached MS

Author Response
We would like to express our sincere gratitude to the reviewer for their thorough and meticulous review of our manuscript. We greatly appreciate the time and effort invested in providing such detailed feedback.
We have carefully considered all observations, corrections, and suggestions made by the reviewer, and have implemented them. All changes from this reviewer have been highlighted in yellow in the final document for easy reference.
Additionally, each of these changes and/or clarifications has been elaborated in detail in the accompanying cover letter. We believe that these revisions have substantially improve the quality of our work.
Comments and Suggestions for Authors
The paper “Response to in vitro micropropagation of plants with different degrees of variegation of the commercial Gymnocalycium cv. Fancy (Cactaceae)” by Carles Cortés-Olmos, Vladimir Marín Guerra-Sandoval, Carla Guijarro-Real, Benito Pineda, Ana Fita, and Adrián Rodríguez-Burruezo, reports the effect of the plant size and color, the explant type, and the cytokinin type over the organogenesis and the variegation of shoots in the in vitro cultured explants.
The objective was not clearly specified in the abstract section, where it was mentioned that the response of the variants of variegated plants has not been investigated in in vitro cultures. However, at the end of the introduction the authors indicated that the objective was to establish an efficient protocol for in vitro propagation of Gymnocalycium cv. Fancy plants with varying degrees of variegation through the evaluation of the effect of type of cytokinin and type of explants from plants with different sizes and variegation proportion, over the organogenic response of the in vitro cultivated explants. Authors also point out that “the aim is to determine the effect of these plant growth regulators on variegated plants and the relationship between the initial plants' variegation proportion and the productivity of shoots with varying degrees of variegation”
Attending reviewer's feedback, we have rewritten the abstract to clearly articulate our aim of establishing an efficient in vitro propagation protocol for Gymnocalycium cv. Fancy, emphasizing the evaluation of cytokinin effects and the relationship between initial variegation, shoot productivity and the final color of these shoots.
Since experimental evidences covering such objectives were reported by the authors in a previous report (A. Micropropagation and acclimatization of Gymnocalycium cv. Fancy (Cactaceae): Developmental responses to different explant types and hormone conditions. Plants 2023, 12, 3932), and the difference with the current study was not presented in the introduction section, and was lightly compared with the results in the manuscript; in addition that due to the ornamental value of the Gymnocalycium gender along with other plant species from the Cactaceae family, there are plenty of reports regarding to the in vitro propagation of these plant species, including the use of different type of explants and use of exogenous cytokinin regulators for activation the development of explants and dormant meristems in the areoles (see for example Micropropagation of cacti—a review. Haseltonia 2014.19 (2014): 46-63, and Method for vegetative propagation of plants. U.S. Patent Application 18/001,732 filed July 27, 2023), the present study lack of originality, and there are not a real novel/new technologic advances in this work.
We appreciate the reviewer's thorough assessment of our study. However, we want to emphasize that our research provides several novel contributions from previous studies:
- This study focuses on variegated plants with diverse ranges of variegation, for which no prior information on in vitro regenerative capacity was available. This unique focus allows us to explore the behavior of variegated Gymnocalycium in vitro and optimizing production protocols for these highly sought-after plants in the ornamental market.
- Our study has been instrumental in identifying which types of explants are most likely to produce shoots with specific variegation levels. This allows for the targeted production of plants that exhibit partial variegation, which are not only capable of developing on their own roots but also possess a highly desirable aesthetic quality for collectors. These results open new opportunities for commercial nurseries to cater to the specialized collector's market, addressing a demand that has not been fully explored in previous studies.
- Furthermore, our study has enabled us to understand how the variegation present in the areoles correlates with the variegation of the obtained shoots. This insight allows for the optimization of production processes.
In this context, we can confirm that our findings have direct practical applications for commercial nurseries, potentially improving both the efficiency and economic viability of variegated cactus production. We believe these points highlight the originality and value of our work, which builds upon existing knowledge to tackle specific aspects of Gymnocalycium propagation that had not been previously explored.
Finally, although the manuscript shows a good understanding of the technical terms, writing and use of statistical analysis, the principal objective mentioned in the introduction section of the manuscript, which was to establish an efficient protocol for in vitro propagation of Gymnocalycium cv. Fancy plants with varying degrees of variegation, was not achieved. As pointed out at the end of the results section, with the results it could be possible to select mixed areolas as starting explants or shoots could be isolated, rooted, and acclimatized, making their ex vitro development to settle down a micropropagation protocol; however such protocol is still pending.
In response to the reviewer's feedback, we have added new sections in both the Materials and Methods and Results chapters detailing the acclimatization process. These additions complete the micropropagation protocol, demonstrating its efficacy from the initial explant selection through to successful ex vitro development. The acclimatization data, although based on a sample of 20 shoots for each variegation category, provides crucial evidence of the protocol's success across different degrees of variegation.
We acknowledge that the initial manuscript may have appeared incomplete without this acclimatization data. The reason for its initial omission was that the majority of the regenerated shoots are being maintained in vitro for further experiments. However, the subset used for acclimatization has allowed us to validate the entire protocol, from in vitro propagation to successful ex vitro establishment.
With these additions, we believe we have addressed the reviewer's concerns and demonstrated that our study has indeed achieved its principal objective of establishing an efficient, complete protocol for the micropropagation of variegated Gymnocalycium cv. Fancy plants. The protocol now encompasses all stages from explant selection to acclimatization.
General comments
Some crucial background information is missing in the introduction section. For example, discussing studies about the diversity in morphologies and colorations of the plant species studied.
We appreciate the reviewer's observation regarding the background information in the introduction section. We acknowledge the importance of providing adequate context. However, we face a particular situation: (1) The specific cultivar used in our study has not been the subject of previous research regarding its morphology and coloration; (2) Furthermore, after an extensive literature review, we have not found studies specifically addressing morphological and coloration diversity in variegated cacti.
Given this scarcity of directly related prior information, we believe our study could be pioneering in this particular aspect.
In the results section, consider using the word frequency instead of the word percentage.
"Frequency" has been substituted for "Percentage" throughout the text where appropriate, in accordance with the reviewer's request.
Improve subtitles by giving more accurate context of the respective contents.
Some subtitles have been modified following the reviewer´s request.
Define early in the text how the frequency or number of activated explants and activated areolas was established. For example, at the beginning of section 2.1. it should be starting by pointing out how the activation of explants were assessed (It was mentioned until 3.4. Experimental design section, row 538)
We appreciate the reviewer's observation. The reason for this placement is that the journal's formatting guidelines require the Materials and Methods section (which includes the experimental design and assessment criteria) to be positioned at the end of the article. However, we included a brief mention of the assessment criteria in section 2.1 as well, to provide readers with this information earlier in the text: “Activated explants were assessed as the number of explants that resulted in some type of response”. This addition would complement the detailed explanation already present in section 3.4.
There are too many confusing and inaccurate sentences, especially in the material and methods section, where also several experimental details are missing.
All details provided by the reviewer regarding the Materials and Methods section have been thoroughly addressed and corrected.
Short explanation of the respective experiment and how the parameter shown were measured showed must be included in the Tables and Figures captions.
As requested by the reviewer, we have modified the captions of several Tables and Figures to include brief explanations of the respective experiments and measurement methods where appropriate.
In some cases, we have maintained the original captions to avoid redundancy with information already presented in the main text. This decision was made to balance the need for clarity with the importance of concise presentation.
Because the experimental design used (Table 11), interaction between factors cannot be evaluated.
We appreciate the reviewer's observation and We would like to clarify our approach and intentions: (1) Our primary aim was not to assess interactions, but rather to investigate how different plant growth regulators affect the various types of explants we are using; (2) Furthermore, the design allowed us to examine the individual effects of regulators on each explant type, providing valuable insights into their specific responses.
Table 5 show the results from the major effect of each factor over the frequency of activated areoles.
Caption has been modified.
The results analyzing the combination of factors (R265-299) and the canonical correlation analysis (R361-428) deserves further explanation along the manuscript. Because the experimental treatments were performed in blocks it is difficult to combine the levels of independent factors such the plant sizes with all the explants type.
Perhaps the use of combination of factors and canonical correlation analysis has not been fully explained and may be the cause of some missunderstanding. It is important to emphasize that the factor combination analysis and the canonical correlation analysis serve distinct purposes in our study: while the former aims to determine which type of explant responds most effectively to a specific treatment, the latter focuses on establishing the relationship between the initial plant coloration, areole pigmentation, and the variegation of the obtained offshoots.
Considering our research objectives, we are not addressing a conventional interaction in the statistical sense. Rather, we are simply checking a combination of factors to determine how each type of explant behaves in the presence of a specific cytokinin with the main aim of identifying the most efficient combinations. This approach allows us to evaluate the specific response of different explant types to various cytokinins, but also determine optimal combinations for shoot induction and proliferation taking into account the differential sensitivity of explant types to cytokinin treatments.
Actually, our experimental design is mainly tailored to elucidate these relationships without necessarily quantifying statistical interactions. This approach aligns with our primary goal of characterizing explant-specific responses to cytokinin treatments, which is crucial for optimizing micropropagation protocols in our study specie.
Also, for canonical correlation analysis the measured variables should come from homogeneous experimental units. So, this condition should be demonstrated for the analyzed treatments. See additional comments in the text.
Taking into account the reviewer's comment regarding the canonical correlation analysis (CCA) and to address this concern, we would like to provide additional information about this analysis:
First of all, we ensured the homogeneity of our experimental units: (a) all plants used in the study were of the same cultivar and grown under identical controlled conditions prior to the experiment (i.e. the same growth conditions); (b) all variables were measured meticulously to ensure consistency (initial size, color of the initial plants, number of areoles …); (c) and experimental units were randomly assigned to treatments to minimize bias.
Besides, to demonstrate the homogeneity and perform a variance check of our experimental units, we conducted Levene's test for each variable across treatments.
Regarding the treatment of non-homogeneous factors, the possible comparison between small and large plants was avoided, and the comparisons focused on evaluating the initial coloration of the plants, the color of activated areoles, and the color of the obtained shoots.
Given the relevance of the observation, the previous explanation about the correlation analysis has been expanded in the materials and methods section: “To ensure the validity of our statistical analyses, we conducted Levene's test to verify the homogeneity of variances across treatments for each variable. Furthermore, to address potential non-homogeneity in plant size, direct comparisons between small and large plants were avoided. Instead, our analyses focused on color-related variables (initial plant coloration, color of activated areoles, and color of obtained shoots) which were consistently measurable across all experimental units regardless of plant size.”
Notice that the last sentence in the results section point out that further research is needed to achieve the principal objective settle down at the beginning of this manuscript.
This observation has already been taken into account in the 'General Comments' section.
Some specific comments
Results
R102 Activated explants were assessed as the number....
The sentence has been modified.
R139 It should be noticed that the terms production and presence or occurrence are used for description of different parameters. In plant cell and tissue culture, production is used for biomass amount.
Taking into account the reviewer's nuance regarding the use of the term "production" when referring to calluses, it has been decided to replace both the subtitle 2.1.1. with "Calli occurrence", and the words in the paragraph with "occurrence".
R240 Efficiency of what? Specify. Also, explain here, and in section 3.4.1, and captions of Tables 8 and 9 how areolar activation was assessed.
With the aim of specifying what areolar activation consists of and detailing how it has been evaluated, an explanatory paragraph has been included in the materials and methods section: "The maximum response capacity of each explant type is determined by the number of areoles present on each (Table 4), as the dormant buds of cacti are contained within these structures. Therefore, in addition to evaluating productivity in terms of the number of calluses and shoots, the efficiency of the explants was analyzed by calculating the percentage of activated areoles. This percentage was calculated by comparing the number of areoles that gave rise to new shoots with respect to the total number of areoles available on each explant. This measure provides a more precise view of the regenerative capacity of each explant type, taking into account its intrinsic potential based on the number of areoles present."
Furthermore, the title of section 2.1.3. “Efficiency related to areolar activation” has been replaced by: “Explant efficiency”; and paragraph “Since the number of areolas available on each type of explant is variable (Table 4), the theoretical response may also differ. In addition to evaluating productivity in number of calluses and number of shoots, the efficiency of the explants was also assessed on the percentage of activated areolas compared to the total number on each explant (Table 5).” has been removed to avoid redundancies (since the specifications are now in the materials and methods section).
R242 … the theoretical response may also differ… The idea is confusing, please explain what theoretical response means.
“Theoretical response” has been replaced by “maximum response capacity” to avoid any kind of confusion. Nevertheless, an explanatory sentence has been included in the materials and methods section that refers to this response capacity and its relationship with the preformed dormant buds contained in the areoles: “The maximum response capacity of each explant type is determined by the number of areoles present on each (Table 4), as the dormant buds of cacti are contained within these structures.”
R244 Explain the criteria used for assessing areolar activation.
Following reviewer instructions, the criteria used for assessing areolar activation has been included in 3.4.1. section: “…this percentage was calculated by comparing the number of areoles that gave rise to new shoots with respect to the total number of areoles available on each explant.”
R245 The idea is confusing. Which results were not maintained? Explain.
Given the reviewer's assessment, it has been decided to simplify the paragraph to make it more dynamic and understandable, avoiding redundancies with the previous section. So: “efficiency result varied some of the results observed in the previous section were maintained, such as the higher efficiency of TDZ1 in areolar activation, the better response of medium and large-sized explants compared to small ones, and the activation ability regardless of the initial variegation degree (except for fully variegated plants) (Table 5).” Has been removed from the text.
R247 Do you mean explants from medium and large sized plants? Explants with different sizes were not described in the material and methods section.
As the reviewer has noted, the correct definition would have been "explants from medium and large sized plants". However, the paragraph has been previously deleted to avoid repeating information and to simplify the text.
R259 Table 5. Is this the cumulative frequency of activated areoles? Explain in the caption the criteria used for assessing areolar activation.
Changes made following the reviewer's instructions
R265- This section deserves further analysis and explanation. Experimental treatments were performed in blocks. Therefore, it is difficult to combine the levels of independent factors such the plant sizes with all the explants type. In fact, explants from small plants are different from the other plant sizes. Usually, combination of factors are used to estimate effects by interaction of such factors, which can not be done in the experimental design used in this work.
As noted by the reviewer, the analysis was necessarily performed in blocks. Our aim was to determine which type of explant-treatment combination could generate a greater organogenic response. To evaluate various alternatives, both small plants and larger plants were included.
The nature of the plant material itself made it impossible to combine some factors. Epicotyls and hypocotyls were obtained from small plants, but these plants were not large enough to obtain perfectly formed apices or central discs containing areoles. Similarly, large plants were too big to consider obtaining epicotyls and hypocotyls. So, small plants (4 to 8 mm) and larger plants (8 to 16 mm) could not be directly compared.
Thus, from each of these starting materials, it was only possible to obtain different types of explants. And these explants were then studied in combination with the growth regulators (BAP8, KIN4, and TDZ1) to determine which protocol could be most appropriate in a commercial production context.
R361 Information about the canonical correlation analysis is needed. Usually, for this statistical procedure the measured variables should come from homogeneous experimental units. So, this condition should be demonstrated for the analyzed treatments.
How the canonical correlations were conducted has been discussed previously.
Material and methods section
R471-474 This sentence should not be in the materials and methods section, and more information about this topic could be provided in the introduction section.
As proposed by the reviewer, the description and characteristics of Gymnocalycium cv. Fancy have been included in the introduction and removed from the materials and methods section.
R489 Induction of what? Please, specify.
The aim of the trial was to stimulate shoot formation through areolar activation. Therefore, 'induction' in this context refers to 'Shoot induction'. The title of the block has been modified to be more specific, as proposed by the reviewer.
R491 How was the degree of coloration assessed? Please, cite Figure 9.
How coloration was assessed has been included in the experimental design section: “Plants were visually classified into four groups depending on the proportion of coloration observed (25%, 50%, 75%, and 100%).” A reference to Figure 9 was also included.
To avoid redundancy, this specification was not repeated in the 'Shoot Induction' section. However, if the reviewer deems it appropriate, we can emphasize this information there as well.
R495 What explants? How the explants to establish the tissue cultures were obtained should be explained here citing Table 11.
A new section was created for this paragraph following the reviewer´s intructions: Section 3.3. Type of explants.
R497 Induction of what? Of the explants? Please explain. Are these this the induction media? Specify.
In order to facilitate the understanding of this fragment, a clarifying sentence has been added: “These induction media were used to stimulate the activation of explants towards shoot production.”
R497-498 Consider exchange by: Besides, a control group of explants was cultivated in the absence of PGRs for each plant size.
The corresponding modification has been made.
R508 Change for by from
The corresponding modification has been made.
R505-512 Consider moving this paragraph to the previous section, perhaps under a subtitle like type of explants
A new section was created for this paragraph following the reviewer´s intructions: Section 3.3. Type of explants.
R513 How was the proportion of coloration settled down? Please stated that 0 color refers to plants fully covered by chlorophyll
The paragraph has been modified in response to the reviewer's requests.
R514 Why was no variegation (0% color) not included in the parenthesis? Please, to refer to the proportion of coloration consider exchanging the numbers in parenthesis by (Figure 9).
The corresponding modification has been made.
R518 ….included as a different control groups…
Green plants (without coloration) were used as a control group to compare their responses with those obtained from variegated plants. These control groups were established to determine the evolution and response of chlorophyllic plants to the same growth regulators and under the same conditions as the explants from colored plants. They were also used to determine their development in the absence of regulators (column Treatment: CONTROL).
R518-520 Please, note and emphasize that treatment with medium sized plants in absence of PGRs is not present in Table 11. Explain why.
Since a larger quantity of medium-sized plants was available, it was decided to evaluate explants from these plants in all media, both in the presence of growth regulators (32 per hormone) and in the absence of regulators (control group, with 64 explants). These data are found in Table 11 and have been highlighted in yellow to facilitate their review.
R521 Table 1. This is a quasi-experimental design, where explants were grouped based on non-random criteria after their plant position, plant size and vegetated proportion, therefore the statistical analysis must be performed by independent groups.
We would like to direct the reviewer's attention to our comprehensive response provided for the comment on line R265.
R532 …five 574 explants… Please revise the text. …five 574… is confusing.
The typo has been corrected
R534 Explants were evaluated or used to evaluate something? Explain what characteristic or response variable was evaluated.
“Evaluated” has been replaced by “cultivated” to be more specific.
R539 consider specify the response variable measured, for example: ..organogenic (shoots and roots appearance) or callogenic (callus tissue appearance)…
The nuances proposed by the reviewer have been included to facilitate understanding of the text.
R560-562. Point out were the results of the relationship are shown.
The correlation table referred to in this sentence has been included in the text following the reviewer's considerations.
Reviewer 2 Report
Comments and Suggestions for Authors
Although the manuscript "Response to in vitro micropropagation of plants with different degrees of variegation of the commercial Gymnocalycium cv. Fancy (Cactaceae)" explores a relevant topic, there is a lack of significant innovation. Most findings corroborate previous studies, which may be a limitation for a favorable review. The discussion could be more in-depth regarding the physiological and biochemical implications of the observed responses, as well as more robust comparisons with previous studies. Some sections, such as the results, present repetitive information, which may compromise the readability and impact of the article. A grammatical and stylistic revision is needed to improve fluency and objectivity, ensuring a more scientific and concise presentation. Finally, the conclusions do not clearly highlight the main contributions to the field of cactus micropropagation, lacking a more assertive synthesis based on the presented data. Below are some revisions:
1. Introduction
a. The introduction does not clearly establish the scientific gap or the specific problem the study aims to address. The text emphasizes the ornamental value of variegated plants but does not clearly present the specific challenges faced in their micropropagation. See: "Variegated plants represent a significant part of the ornamental plant market due to their aesthetic appearance."
b. Broad statements without direct references may compromise the scientific credibility of the text. Readers may question the validity of this information without support from updated sources. See: "Nowadays, the presence of a wide range of variegated plants, including popular ones like pothos, alocasias, and monsteras, reflects their significance in gardening and landscaping."
c. The sentence is long and contains redundant information, such as the repeated explanation of the relationship between vigor and variegation. This compromises the objectivity of the introduction. See: "Therefore, although these plants tend to be less vigorous than the non-variegated ones, due to their lower amount of chlorophyll, which affects their photosynthetic capacity, it is not surprising that commercial nurseries work intensively to obtain plants with new colorations or patterns to annually introduce into the market."
d. The transition between the topic of variegation and the grafting technique is not well-developed, resulting in an abrupt shift in subject matter. Readers may struggle to understand the relationship between grafting and in vitro micropropagation. See: "The grafting technique is widely used in the mass propagation of cacti due to its advantages against plants grown from their roots."
e. The introduction lacks a clear and objective formulation of the study's purpose and hypotheses. Readers need to understand the main objectives precisely. See: "Thus, the study focuses on the in vitro response of Gymnocalycium plants with varying degrees of variegation." A possible revision could be: "The present study aims to establish an efficient in vitro propagation protocol for Gymnocalycium cv. Fancy, assessing the influence of different variegation levels on shoot formation and growth regulator responses."
f. The use of connectors like "so" to start a new clause is inappropriate for formal scientific texts, compromising fluency. See: "Completely achlorophyllous plants must remain grafted due to their inability to photosynthesize, so they are generally unpopular among collectors."
g. The lack of direct references to previous studies that demonstrate this gap may weaken the study's justification. See: "Therefore, there is a lack of information regarding many ornamental species that could have a significant impact on the market."
2. Results and Discussion
a. The presentation of results combines multiple pieces of information into a single sentence, making it dense to read. Additionally, the section lacks a clearer structure to highlight the findings in an organized manner. See: "The number of explants that resulted in some type of response (whether it be the formation of shoots, callus, or both), as well as their proportion relative to the total number of explants included for each factor and evaluated variable, are shown in Table 1."
b. Results are presented repeatedly throughout the text, compromising objectivity. The unnecessary repetition of values mentioned in tables should be avoided. See: "It is observed that the presence of TDZ1 in the medium significantly favored the activation of explants, with a response rate of 83.11%, while the other treatments (BAP8, 60.53%; KIN4, 50.67%) showed a lower activation efficiency, but slightly higher than those observed in the control group (46.77%) (Table 1)." It is advisable to present the main findings concisely and refer to the table for details.
c. The discussion is too descriptive and lacks an in-depth critical analysis. There is no detailed comparison with previous studies to validate or contrast the findings. See: "These findings suggest that hypocotyl and epicotyl explants, composed of younger and less mature tissues, are not so efficient in activating their areoles with the only use of cytokinins." It is beneficial to include a comparison with relevant literature and discuss possible reasons for the observed differences.
d. The term "considerably worse" is vague and imprecise for a scientific text, requiring a more quantitative and objective explanation. See: "Completely variegated explants responded considerably worse to treatments compared to plants with chlorophyll tissues."
e. The statement lacks quantitative and statistical support to justify the conclusion of "repeatability." A variability or reproducibility analysis should be included. See: "Thus, the repeatability of the response is confirmed in this trial."
f. The lack of smooth transitions between results and discussion hinders the reading flow, making the section fragmented. See: "On the other hand, the number of calluses formed was influenced by plant size, with larger plants showing better responses."
g. The conclusion is made without complementary ex vitro acclimatization tests or comparisons with alternative protocols. See: "This study confirms the feasibility of using central discs for efficient propagation of variegated Gymnocalycium."
h. The analysis of figures and tables does not fully explore the observed trends, limiting the interpretation of visual data. See: "Figure 3 shows the interaction between hormone treatment and explant type on shoot production."
i. The section lacks a more in-depth discussion on how the findings can be commercially applied. See: "These results could be applicable to other cactus species that may attract interest from consumers and collectors."
j. The section ends in a generic manner, without a clear recap of the main findings. See: "Therefore, in vitro micropropagation can be considered a useful tool for the propagation of Gymnocalycium cv. Fancy."
3. Methodology
a. There is no information regarding light intensity (in µmol m⁻² s⁻¹), type of lamps (LED, fluorescent), etc. See: "Seedlings developed under in vitro conditions inside a growth room at 26±2ºC on shelves with a 16 h light / 8 h dark photoperiod."
b. The term "domestic bleach solution" is inappropriate for scientific publications as it does not specify the exact concentration of sodium hypochlorite. Additionally, rinsing details are missing. See: "Seeds were treated under aseptic conditions in a laminar flow cabinet for 1 min in 70% ethanol, followed by 25 min in 15% domestic bleach solution supplemented with 0.08% of the surfactant Tween-20."
c. There is no mention of the number of replicates, total number of samples per group, or how treatments were randomized, compromising the statistical validity of the experiment. This is a critical issue.
d. The description of explants is vague, lacking details on exact size, position in the plant, and selection criteria, which hinders study replicability. See: "Different types of explants (apices, central discs, epicotyls, and hypocotyls) were obtained from plants."
e. There is no mention of negative controls (without regulators) or positive controls (previously established treatments), which may hinder the interpretation of results.
g. There is no information regarding the statistical tests used, the significance level adopted, and the software Conclusion
a. The extrapolation to other cactus species is made without direct experimental support, which may compromise the credibility of the conclusion. See: "This protocol could be applied to other ornamental cacti, expanding propagation options for the industry."
b. There is no mention of the study's limitations, such as potential challenges in ex vitro acclimatization, genetic variation of explants, or the need for optimization for different cultivars. See: "Overall, the findings provide valuable insights into the micropropagation of variegated ornamental cacti."
c. The use of subjective terms such as "interesting insights" is not appropriate for a high-impact scientific conclusion, as it does not convey precision. See: "This study provides interesting insights into the behavior of variegated plants in vitro."used. See: "Data were analyzed statistically to determine significant differences."
4. Conclusion
a. The extrapolation to other cactus species is made without direct experimental support, which may compromise the credibility of the conclusion. See: "This protocol could be applied to other ornamental cacti, expanding propagation options for the industry."
b. There is no mention of the study's limitations, such as potential challenges in ex vitro acclimatization, genetic variation of explants, or the need for optimization for different cultivars. See: "Overall, the findings provide valuable insights into the micropropagation of variegated ornamental cacti."
c. The use of subjective terms such as "interesting insights" is not appropriate for a high-impact scientific conclusion, as it does not convey precision. See: "This study provides interesting insights into the behavior of variegated plants in vitro."
Author Response
We would like to express our sincere gratitude to the reviewer for their thorough and meticulous review of our manuscript. We greatly appreciate the time and effort invested in providing such detailed feedback.
We have carefully considered all observations, corrections, and suggestions made by the reviewer, and have implemented them All changes from this reviewer have been highlighted in blue in the final document for easy reference.
Additionally, each of these changes and/or clarifications has been elaborated in detail in the accompanying cover letter. We believe that these revisions have substantially improve the quality of our work.
- Introduction
- The introduction does not clearly establish the scientific gap or the specific problem the study aims to address. The text emphasizes the ornamental value of variegated plants but does not clearly present the specific challenges faced in their micropropagation. See: "Variegated plants represent a significant part of the ornamental plant market due to their aesthetic appearance."
- Broad statements without direct references may compromise the scientific credibility of the text. Readers may question the validity of this information without support from updated sources. See: "Nowadays, the presence of a wide range of variegated plants, including popular ones like pothos, alocasias, and monsteras, reflects their significance in gardening and landscaping."
- The sentence is long and contains redundant information, such as the repeated explanation of the relationship between vigor and variegation. This compromises the objectivity of the introduction. See: "Therefore, although these plants tend to be less vigorous than the non-variegated ones, due to their lower amount of chlorophyll, which affects their photosynthetic capacity, it is not surprising that commercial nurseries work intensively to obtain plants with new colorations or patterns to annually introduce into the market."
- The transition between the topic of variegation and the grafting technique is not well-developed, resulting in an abrupt shift in subject matter. Readers may struggle to understand the relationship between grafting and in vitro micropropagation. See: "The grafting technique is widely used in the mass propagation of cacti due to its advantages against plants grown from their roots."
- The introduction lacks a clear and objective formulation of the study's purpose and hypotheses. Readers need to understand the main objectives precisely. See: "Thus, the study focuses on the in vitro response of Gymnocalycium plants with varying degrees of variegation." A possible revision could be: "The present study aims to establish an efficient in vitro propagation protocol for Gymnocalycium cv. Fancy, assessing the influence of different variegation levels on shoot formation and growth regulator responses."
- The use of connectors like "so" to start a new clause is inappropriate for formal scientific texts, compromising fluency. See: "Completely achlorophyllous plants must remain grafted due to their inability to photosynthesize, so they are generally unpopular among collectors."
- The lack of direct references to previous studies that demonstrate this gap may weaken the study's justification. See: "Therefore, there is a lack of information regarding many ornamental species that could have a significant impact on the market."
Following the reviewer's comments, we have carefully addressed all the points raised regarding the introduction section.
First of all, we clarified the scientific gap and the specific problem our study aims to address: "there is a significant gap in knowledge regarding many ornamental species, particularly when propagating variegated individuals or those with particular color patterns." We have corrected and removed broad statements lacking direct references to maintain scientific credibility and, additionally, we aimed to simplify long sentences and eliminate redundant information to enhance clarity and objectivity.
We improved the transition between topics, particularly between the variegation and its relationship with grafting techniques. A more objective formulation of the study's purpose and objectives have been provided: "The present study aims to establish an efficient in vitro propagation protocol for Gymnocalycium cv. Fancy, assessing the influence of different variegation levels on shoot formation and growth regulator responses." We also added a crucial point about the potential of in vitro propagation to produce variegated plants that can grow on their own roots, addressing collectors' preferences and market demands.
Furthermore, the text and language have been thoroughly revised, so we believe these changes have significantly improved the introduction, providing a clearer focus and objective for our research.
- Results and Discussion
- The presentation of results combines multiple pieces of information into a single sentence, making it dense to read. Additionally, the section lacks a clearer structure to highlight the findings in an organized manner. See: "The number of explants that resulted in some type of response (whether it be the formation of shoots, callus, or both), as well as their proportion relative to the total number of explants included for each factor and evaluated variable, are shown in Table 1."
To avoid excessive use of independent tables, we deliberately chose to use a single comprehensive table that combines all results related to explant activation, both in number and percentage. This approach was intended to provide a concise overview of the data.
However, we understand the reviewer's concern about the density of information so, if the reviewer has any specific suggestions on how to improve the presentation of these results, we would be more than willing to address their recommendations.
- Results are presented repeatedly throughout the text, compromising objectivity. The unnecessary repetition of values mentioned in tables should be avoided. See: "It is observed that the presence of TDZ1 in the medium significantly favored the activation of explants, with a response rate of 83.11%, while the other treatments (BAP8, 60.53%; KIN4, 50.67%) showed a lower activation efficiency, but slightly higher than those observed in the control group (46.77%) (Table 1)." It is advisable to present the main findings concisely and refer to the table for details.
In response to the reviewer's observation, we have removed redundant numerical data from the text and direct readers to the corresponding tables for detailed results. This modification simplifies the reading process.
- The discussion is too descriptive and lacks an in-depth critical analysis. There is no detailed comparison with previous studies to validate or contrast the findings. See: "These findings suggest that hypocotyl and epicotyl explants, composed of younger and less mature tissues, are not so efficient in activating their areoles with the only use of cytokinins." It is beneficial to include a comparison with relevant literature and discuss possible reasons for the observed differences.
Considering the reviewer's comment, we would like to emphasize that the Studies focusing on regenerative responses usually present an inherent descriptive nature. This is essential for accurately reporting and analyzing the observed phenomena, especially in novel research areas.
Moreover, it is important to note that the topic addressed in our study is quite new, with limited available literature for direct comparison. Despite this challenge, we have made a concerted effort to enrich our work with relevant studies and reports, incorporating a total of 60 references throughout the manuscript: (i) 25 references in the introduction ; (2) 33 references in the results and discussion sections and ; (3) 2 references in the materials and methods section.
Specifically, in the paragraph mentioned by the reviewer, we have included 9 citations [26-34] to provide context and support for our findings. These references, while not always directly comparable, offer valuable insights and help situate our work within the cacti regeneration studies.
- The term "considerably worse" is vague and imprecise for a scientific text, requiring a more quantitative and objective explanation. See: "Completely variegated explants responded considerably worse to treatments compared to plants with chlorophyll tissues."
Following the reviewer´s observation regarding the use of imprecise language, we have removed the word "considerably" from the sentence in question, avoiding subjective observations and focusing on the quantitative differences observed between the variegated and chlorophyll-containing explants.
- The statement lacks quantitative and statistical support to justify the conclusion of "repeatability." A variability or reproducibility analysis should be included. See: "Thus, the repeatability of the response is confirmed in this trial."
We have revised the statement to avoid implying a formal repeatability analysis. The sentence has been modified to: "Thus, similar results have been obtained in this study, consistent with previous findings."
- The lack of smooth transitions between results and discussion hinders the reading flow, making the section fragmented. See: "On the other hand, the number of calluses formed was influenced by plant size, with larger plants showing better responses."
We have considered this comment. We hope that new with the in depth writting changes made in the manuscript, this issue has improved.
- The conclusion is made without complementary ex vitro acclimatization tests or comparisons with alternative protocols. See: "This study confirms the feasibility of using central discs for efficient propagation of variegated Gymnocalycium."
In response to the reviewer's feedback, we have added new sections in both the Materials and Methods and Results chapters detailing the acclimatization process. These additions complete the micropropagation protocol, demonstrating its efficacy from the initial explant selection through to successful ex vitro development. The acclimatization data, although based on a sample of 20 shoots for each variegation category, provides crucial evidence of the protocol's success across different degrees of variegation.
We acknowledge that the initial manuscript may have appeared incomplete without this acclimatization data. The reason for its initial omission was that the majority of the regenerated shoots are being maintained in vitro for further experiments. However, the subset used for acclimatization has allowed us to validate the entire protocol, from in vitro propagation to successful ex vitro establishment.
With these additions, we believe we have addressed the reviewer's concerns and demonstrated that our study has indeed achieved its principal objective of establishing an efficient, complete protocol for the micropropagation of variegated Gymnocalycium cv. Fancy plants. The protocol now encompasses all stages from explant selection to acclimatization.
- The analysis of figures and tables does not fully explore the observed trends, limiting the interpretation of visual data. See: "Figure 3 shows the interaction between hormone treatment and explant type on shoot production."
Our approach in presenting the results has been to focus on the most relevant and significant findings, avoiding redundancy and less pertinent information to enhance overall comprehension. The aim was to highlight key outcomes that directly address the study's objectives while maintaining clarity and conciseness in the manuscript.
We have intentionally avoided an overly detailed analysis to prevent overwhelming the reader with information that is not be central to the main findings of the study. This approach allows for a more focused discussion on the most interesting results.
However, we understand the importance of thorough data interpretation. Therefore, we have introduced a new paragraph to highlight some of the extractable results from Figure 3: “Regarding the responses observed in explants from small plants, epicotyl explants showed a lower response compared to hypocotyls, both in terms of absolute production (Figure 3a) and the frequency of areole activation (Figure 3b). These results could be linked to the higher callus production observed in epicotyls throughout the culture period (Table 2).”
- The section lacks a more in-depth discussion on how the findings can be commercially applied. See: "These results could be applicable to other cactus species that may attract interest from consumers and collectors."
Following reviewer's suggestion we have added a new paragraph (“Acclimatization of the obtained shoots” section) that elaborates on how our results can be relevant and potentially extrapolated to a commercial level. This addition discusses the protocol's reliability in producing variegated plants, its potential application to other cacti species of interest to collectors, and its significance for the ornamental plant industry. “These results evidence that the protocol can reliably produce a range of variegated plants with high survival rates during the critical acclimatization phase. Moreover, the methodology developed could potentially be extrapolated to other variegated cacti species of interest to collectors. The ability to consistently produce variegated plants on their own roots not only meets the expectations of collectors but also opens up a new market niche associated with colored cacti. This approach could significantly impact the ornamental plant industry by providing a reliable method for mass-producing variegated specimens, thus meeting the growing demand for these cacti varieties in the market.”
- The section ends in a generic manner, without a clear recap of the main findings. See: "Therefore, in vitro micropropagation can be considered a useful tool for the propagation of Gymnocalycium cv. Fancy."
We appreciate the reviewer's observation regarding the conclusion of the Results and Discussion. We tried to addres this concern by adding a new subsection at the end of the Results and Discussion section: "Acclimatization of the obtained shoots". This new subsection serves to conclude the discussion in a more specific manner, highlighting the relevance of the described protocol and its commercial utility.
- Methodology
- There is no information regarding light intensity (in µmol m⁻² s⁻¹), type of lamps (LED, fluorescent), etc. See: "Seedlings developed under in vitro conditions inside a growth room at 26±2ºC on shelves with a 16 h light / 8 h dark photoperiod."
As requested by the reviewer, we have included the type of lamp used: LED type. The light intensity was defined in the text: “photosynthetic photon flux of 50 μmol m⁻² s⁻¹”.
- The term "domestic bleach solution" is inappropriate for scientific publications as it does not specify the exact concentration of sodium hypochlorite. Additionally, rinsing details are missing. See: "Seeds were treated under aseptic conditions in a laminar flow cabinet for 1 min in 70% ethanol, followed by 25 min in 15% domestic bleach solution supplemented with 0.08% of the surfactant Tween-20."
We appreciate the reviewer's attention to detail. We would like to clarify that the concentration of sodium hypochlorite is indeed specified in the text as 4% (v/v). However, we understand that this information might have been overlooked.
Regarding the rinsing details, we have now expanded on the information provided. While it was previously mentioned that the seeds were rinsed 3 times with distilled water, we have now added that each rinse lasted for 5 minutes. This addition provides a more comprehensive description of the sterilization process.
The revised text now reads: “For their disinfection, seeds were treated under aseptic conditions in a laminar flow cabinet (model AH-100, Telstar, Terrassa, Spain) for 1 min in 70% ethanol (v/v), continued by 25 min in 15% domestic bleach solution (v/v; 4% sodium hypochlorite) supplemented with 0.08% of the surfactant Tween-20 (v/v). Finally, seeds were rinsed 3 times in distilled sterilized water, with each rinse lasting 5 minutes, before sowing.”
- There is no mention of the number of replicates, total number of samples per group, or how treatments were randomized, compromising the statistical validity of the experiment. This is a critical issue.
We appreciate the reviewer's observation regarding the experimental design details. We would like to clarify that the information on the number of replicates and the total number of samples per group is detailed in Table 11 of section "3.4 Experimental Design". However, if reviewer thinks thaty more details are necessary, we will be willing to include more.
Regarding the randomization of treatments, we acknowledge that this information was not explicitly included in the original text. To address this omission, we have added the following sentence to section 3.4: "Explants were assigned to different treatments using a random number system, ensuring a random distribution of samples among the experimental groups."
- The description of explants is vague, lacking details on exact size, position in the plant, and selection criteria, which hinders study replicability. See: "Different types of explants (apices, central discs, epicotyls, and hypocotyls) were obtained from plants."
In response to the reviewers' request, we have created a new section to appropriately describe the explants: 3.3. Type of Explants. This section includes the details requested by the reviewer 2 to facilitate understanding about the obtainment, size, and morphology of the explants, as well as their positioning in the culture medium. This addition enhances the clarity and reproducibility of our experimental methods.
- There is no mention of negative controls (without regulators) or positive controls (previously established treatments), which may hinder the interpretation of results.
This information is now located in section 3.5. Experimental Design: “Furthermore, green plants (0% color) were evaluated as a different control groups: (a) on one hand, medium-sized plants subjected to the presence of PGRs and, (b) on the other hand, plants of all evaluated sizes (small, medium and large-sized) in absence of PGRs (Table 11).”
- There is no information regarding the statistical tests used, the significance level adopted, and the software
The information requested by the reviewer is now located in section “3.6. Statistical Analysis” of the article. This section describes the statistical analyses performed and the significance level adopted for the analyses. However, if the reviewer requires any clarification or believes that the information needs to be clarified at any point, we are willing to do so.
- Conclusion
- The extrapolation to other cactus species is made without direct experimental support, which may compromise the credibility of the conclusion. See: "This protocol could be applied to other ornamental cacti, expanding propagation options for the industry."
- There is no mention of the study's limitations, such as potential challenges in ex vitro acclimatization, genetic variation of explants, or the need for optimization for different cultivars. See: "Overall, the findings provide valuable insights into the micropropagation of variegated ornamental cacti."
- The use of subjective terms such as "interesting insights" is not appropriate for a high-impact scientific conclusion, as it does not convey precision. See: "This study provides interesting insights into the behavior of variegated plants in vitro."
In response to the reviewer's observations, we have revised the conclusion to address the points raised. The extrapolation to other cactus species has been moderated, the study's limitations have been acknowledged, and subjective terms have been replaced with more precise language.
Reviewer 3 Report
Comments and Suggestions for Authors
The manuscript reports on the micropropagation of variegated ornamental plants of Gymnocalycium cv. Fancy with the aim of establishing protocols for in vitro propagation of plants having different extent of colour variegation. By carefully investigating different starting explants using specific plant growth regulators, significant percentages of variegated shoots were successfully obtained and the protocol optimized. The manuscript is quite well written and experimental work well conducted.
Minor hints:
Table headings should be more detailed and self-explanatory
Figures 7-10: Figure legends should be improved by including the plant species
Table 11: Small (4-12mm) should be (4-8mm)?
Minor English revision suggested.
Author Response
We sincerely thank the reviewer for their positive evaluation and constructive observations. We appreciate the time and effort invested in reviewing our manuscript.
The reviewer's feedback on the introduction, results, discussion, methodology, and conclusions is greatly valued. We are grateful for the recognition of our work's potential practical applications in commercial propagation.
We have taken note of the suggestion to include more recent bibliographic references and have addressed this in our revisions. We have incorporated three additional citations from 2023 and 2024 (highlighted in red) and we hope these new references complement the current state presented in our manuscript.
Lopez-Granero, M.; Arana, A.; Regalado, J.J.; Encina, C.L. Mass micropropagation and in vitro fowering of Mammillaria vetula ssp. gracilis var. Arizonica Snowcap. PCTOC 2023, 155, 759–771. doi: 10.1007/s11240-023-02597-1.
Manokari, M.; Faisal, M.; Alatar, A.A.; Shekhawat, M.S. Optimization of in vitro regeneration of Ferocactus peninsulae (Barrel Cactus) through transverse thin cell layer (tTCL) culture: a strategy for large‑scale propagation. PCTOC 2024, 159:48. Doi: 10.1007/s11240-024-02910-6.
Oo, K.T.; Lynn, Z.M.; Oo, K.Z.; Htwe, M.Y.; Htet, W.T.; Soe, W.W.; Tun, W. In vitro Propagation of Three Pitaya Varieties (Hylocereus undatus, Hylocereus polyrhizus and Hylocereus megalanthus) with the Use of Different BAP Concentrations. JSIR 2023, 12(2), 33-39. Doi: 10.31254/jsir.
Once again, we thank the reviewer for their thorough and encouraging review.
Reviewer 4 Report
Comments and Suggestions for Authors
The work is certainly of interest and could also have practical applications in the commercial propagation of the species under study in the medium term.
The introduction appropriately presents the state of the art, providing an in-depth analysis of the topic. It might be advisable to include more recent bibliographic references.
The presentation of the results and the discussion are well-developed, comparing the obtained data with existing studies on related species. Both the propagation process in a strict sense, the color analysis, and the rooting phase are correctly and comprehensively addressed.
The methodology is clearly described and complete.
The conclusions provide a clear and concise summary of the study
Author Response
We sincerely appreciate your encouraging and motivating feedback. Your recognition of our work and how we have presented our findings is gratifying. We are delighted that you find our study promising and consider our manuscript suitable for publication in its present state. Your positive remarks inspire us to explore and deepen our research in this field further.
Round 2
Reviewer 2 Report
Comments and Suggestions for Authors
The manuscript presents a relevant approach for the in vitro micropropagation of Gymnocalycium cv. Fancy, with a well-structured experimental design and results that contribute to the scientific literature on ornamental cacti. However, there are points that need to be improved before final acceptance. Problems with fluidity in the writing, inconsistent use of technical terminology, inadequate grammatical constructions and excessive repetition of terms were identified, which compromise the clarity and objectivity of the text. In addition, there are inconsistencies in the presentation of growth regulators, as well as in the structuring of some sentences that impact comprehension. Given that these adjustments are essential to ensure an adequate level of scientific writing, I recommend that the manuscript be revised before final acceptance.
Comments on the Quality of English Language- Example: "This study is aimed to establish efficient in vitro propagation protocols for Gymnocalycium cv. Fancy..."
Needed correction: "This study aims to establish efficient in vitro propagation protocols for Gymnocalycium cv. Fancy..."
Justification: "Is aimed to" is not a natural construction. The correct form in academic English would be "aims to."
- Example: "Central discs treated with 1µM TDZ gave the best shoot production results."
Needed correction: "The best shoot production results were obtained with central discs treated with 1µM TDZ."
Justification: The sentence needs to be reformulated to improve the formality and clarity of the information.
- The term "variegation" is used throughout the text, but sometimes seems misplaced in certain sentences. For example:
Example: "Thus, the study focuses on the in vitro response of Gymnocalycium plants with varying degrees of variegation (Figure 2). Therefore, the objective of the experiment is to establish an efficient protocol for in vitro propagation of Gymnocalycium cv. Fancy plants with varying degrees of variegation."
Needed correction: The second sentence could be reworded to avoid excessive repetition: "Therefore, this study aims to establish an efficient protocol for the in vitro propagation of Gymnocalycium cv. Fancy with different levels of variegation."
Rationale: Avoids redundancy and improves clarity.
- The term "Gymnocalycium cv. Fancy" appears repeatedly and unnecessarily. In nearby passages, it could be replaced by "this cultivar" or "this species" to avoid excessive repetition and make reading more fluid.
- In some parts of the text, 6-Benzylaminopurine is written as BAP and in others it appears in full. There should be uniformity throughout the text.
- Example: "The presence of TDZ1 in the medium had a highly detrimental effect on root formation (with root emergence being minimal), while the lack of cytokinins favored root emission."
Needed correction: "The presence of TDZ1 in the medium had a highly detrimental effect on root formation, minimizing root emergence, while the absence of cytokinins promoted root development."
Justification: The part in parentheses could be incorporated into the sentence for greater fluidity.
Author Response
Dear Reviewer,
We would like to express our sincere gratitude for the time and effort you have dedicated to reviewing our manuscript for a second time. Your comments and detailed observations have been fundamental to improving our work's quality.
We have carefully addressed all the points raised in your review in this revised version. So we hope these revisions have substantially enhanced the manuscript and now meet the reviewer´s expectations for publication. All changes have been highlighted in purple in the final document for easy reference.
Once again, thank you for your dedication to ensuring the quality of this submission. We look forward to your feedback and hope this version is suitable for final acceptance.
Comments and Suggestions for Authors
The manuscript presents a relevant approach for the in vitro micropropagation of Gymnocalycium cv. Fancy, with a well-structured experimental design and results that contribute to the scientific literature on ornamental cacti. However, there are points that need to be improved before final acceptance. Problems with fluidity in the writing, inconsistent use of technical terminology, inadequate grammatical constructions and excessive repetition of terms were identified, which compromise the clarity and objectivity of the text. In addition, there are inconsistencies in the presentation of growth regulators, as well as in the structuring of some sentences that impact comprehension. Given that these adjustments are essential to ensure an adequate level of scientific writing, I recommend that the manuscript be revised before final acceptance.
Comments on the Quality of English Language
- Example: "This study is aimed to establish efficient in vitro propagation protocols for Gymnocalycium cv. Fancy..."
Needed correction: "This study aims to establish efficient in vitro propagation protocols for Gymnocalycium cv. Fancy..."
Justification: "Is aimed to" is not a natural construction. The correct form in academic English would be "aims to."
Following the reviewer’s recommendation, the suggested modification has been made, and the text now reads: "This study aims to establish efficient in vitro propagation protocols for Gymnocalycium cv. Fancy...".
- Example: "Central discs treated with 1µM TDZ gave the best shoot production results."
Needed correction: "The best shoot production results were obtained with central discs treated with 1µM TDZ."
Justification: The sentence needs to be reformulated to improve the formality and clarity of the information.
In order to improve the formality and clarity of the information, the sentence has been replaced as suggested by the reviewer and now reads: "The best shoot production results were obtained with central discs treated with 1µM TDZ".
- The term "variegation" is used throughout the text, but sometimes seems misplaced in certain sentences. For example:
Example: "Thus, the study focuses on the in vitro response of Gymnocalycium plants with varying degrees of variegation (Figure 2). Therefore, the objective of the experiment is to establish an efficient protocol for in vitro propagation of Gymnocalycium cv. Fancy plants with varying degrees of variegation."
Needed correction: The second sentence could be reworded to avoid excessive repetition: "Therefore, this study aims to establish an efficient protocol for the in vitro propagation of Gymnocalycium cv. Fancy with different levels of variegation."
To improve clarity, the sentence has been corrected as suggested by the reviewer and now reads: "Therefore, this study aims to establish an efficient protocol for the in vitro propagation of Gymnocalycium cv. Fancy with different levels of variegation".
Rationale: Avoids redundancy and improves clarity.
- The term "Gymnocalycium cv. Fancy" appears repeatedly and unnecessarily. In nearby passages, it could be replaced by "this cultivar" or "this species" to avoid excessive repetition and make reading more fluid.
Following the reviewer’s instructions, the term "Gymnocalycium cv. Fancy" has been reviewed and replaced throughout the manuscript to improve readability. As a result, this term now appears only 8 times in total (excluding the title, abstract, and bibliography) and does not repeat within any single section. In other instances, we have referred to it as "form" or "cultivar" to avoid redundancy.
- In some parts of the text, 6-Benzylaminopurine is written as BAP and in others it appears in full. There should be uniformity throughout the text.
The full name "6-Benzylaminopurine" (BAP), as well as the other hormones mentioned (KIN and TDZ), has been retained in the headings and captions of tables and figures to ensure clarity and facilitate a comprehensive understanding of these elements.
Additionally, in the Materials and Methods section, the full names have been included once again with the aim of providing the reader with all necessary information to understand how the experiment was conducted without needing to refer back to the abstract. However, if the reviewer considers it more appropriate, we are willing to replace the full names with their respective abbreviations in this section. We have no evidence that the extended terms appear again in the text.
- Example: "The presence of TDZ1 in the medium had a highly detrimental effect on root formation (with root emergence being minimal), while the lack of cytokinins favored root emission."
Needed correction: "The presence of TDZ1 in the medium had a highly detrimental effect on root formation, minimizing root emergence, while the absence of cytokinins promoted root development."
Justification: The part in parentheses could be incorporated into the sentence for greater fluidity.
The sentence has been modified following the reviewer’s suggestion and now reads: "The presence of TDZ1 in the medium had a highly detrimental effect on root formation, minimizing root emergence, while the absence of cytokinins promoted root development".